# SinI and SinR function differently in biofilm formation, rhizosphere colonization, and biocontrol efficacy between *Bacillus velezensis* and *B. subtilis*

Zhibo Wang,[1] Rui Wang,[2] Shilei Jiang,[1] Yuqing Zheng,[1] Qiankun Jiang,[1] Li Wang,[1] Jun Tan,[2] Xiuyun Zhao,[1] Gaofu Qi[1]

**ABSTRACT** Numerous *Bacillus* species, in particular *B. subtilis* and *B. velezensis*, are usually used as effective biocontrol agents against plant diseases, leveraging their ability to form biofilms for robust colonization of the rhizosphere. In *B. subtilis*, SinI positively influences biofilm formation, rhizosphere colonization, and biocontrol efficacy, whereas SinR has a negative impact. To boost the biocontrol efficacy of *B. velezensis* R9 against tobacco bacterial wilt, we engineered the deletion of *sinI* and *sinR* genes in this strain, respectively. Contrary to expectations, deleting *sinR* impaired biofilm formation, rhizosphere colonization, plant resistance induction, and bacterial wilt control. Conversely, the R9Δ*sinI* strain showed notably enhanced biofilm, colonization, and biocontrol efficacy relative to both R9 and R9Δ*sinR* strains. Complementing R9Δ*sinI* with *sinI* and R9Δ*sinR* with *sinR* confirmed that SinI negatively and SinR positively regulate biofilm formation in R9, regardless of originating from *B. velezensis* or *B. subtilis*. By contrast, *sinI* knockout in *B. subtilis* M6 caused a marked decline in biofilm formation but could be partially reversed by complementary expression of *sinI* whether it was from *B. subtilis* or *B. velezensis*. Conversely, *sinR* knockout in M6 sharply decreased biofilm formation. In summary, SinI negatively and SinR positively regulate biofilm formation in *B. velezensis*, contrasting with their roles in *B. subtilis*. Consequently, deleting *sinI*, not *sinR*, in *B. velezensis* enhances biofilm formation, promoting root colonization, plant resistance, and disease control.

**IMPORTANCE** *Bacillus* species, exemplified by *B. subtilis* as a model organism for Gram-positive bacteria, have been extensively studied, particularly regarding biofilm formation. Biofilms represent a form of quorum sensing in microbial communities, and the biocontrol efficacy of *Bacillus* species in the rhizosphere, against plant pathogens, hinges on their biofilm-forming capabilities. In *B. subtilis*, the regulatory proteins SinI and SinR are known to have opposing functions in biofilm formation, with SinI facilitating and SinR inhibiting biofilm development. Drawing from this foundational knowledge, we endeavored to knock out the *sinR* gene in *B. velezensis*, a biocontrol bacterium, to enhance biofilm formation and, consequently, its colonization of the rhizosphere and biocontrol efficacy. Unexpectedly, the deletion of *sinR* reduced the bacterium's proficiency in biofilm formation and its ability to colonize the rhizosphere, resulting in a decrease in biocontrol effectiveness. On the contrary, the knockout of *sinI* promoted biofilm formation, bolstered the strain's colonization capacity in the rhizosphere, and thus fortified its biocontrol efficacy. These findings underscore that SinI and SinR exert divergent, even antithetical effects in *Bacillus* species. Insights gleaned from *B. subtilis* research cannot be extrapolated to encompass all *Bacillus* species, at least not *B. velezensis*, indicating the need for species-specific investigations.

**KEYWORDS** *Bacillus velezensis*, *sinI*, *sinR*, biofilm, colonization, bacterial wilt disease

Address correspondence to Gaofu Qi, qigaofu@mail.hzau.edu.cn.

The authors declare no conflict of interest.

*B*acillus species, such as *B. velezensis* and *B. subtilis*, are Gram-positive, spore-forming bacteria prevalent in plant rhizospheres. Known as plant growth-promoting rhizobacteria (PGPRs) (1), they suppress soil pathogens and foster plant growth by antibiosis, inducing systemic resistance and bolstering tolerance to environmental stresses (2–4). Biofilm formation is pivotal for PGPRs' rhizosphere colonization, creating protective multicellular communities embedded in an extracellular matrix of exopolysaccharides (EPS), proteins, and poly-γ-glutamic acid (γ-PGA) (2–4). This structure fortifies bacteria against stresses and links robust biofilm capability on roots to enhanced disease control by PGPRs (5, 6).

Biofilm formation is a quorum sensing (QS) response observed in many microorganisms, including species of *Bacillus*. When environmental conditions become unfavorable for cell growth, such as nutrient depletion or the accumulation of toxic metabolites, the cells initiate a QS response. In many *Bacillus* species, this involves the production of cyclic lipopeptides like surfactin, which act as signaling molecules to activate histidine kinases (e.g., KinA, KinB, KinC, KinD, and KinE). These kinases subsequently transfer phosphate groups to Spo0F and Spo0B, which then relay the phosphate to the master regulator Spo0A. Phosphorylated Spo0A regulates the expression of biofilm-related genes (7, 8). Spo0F and Spo0B function as phosphate transfer proteins within this signaling cascade, while Spo0A serves as a global regulator that influences various metabolic pathways in *Bacillus*, including biofilm formation (9). In *B. subtilis*, biofilm formation is orchestrated by regulators like SinR and SinI. SinR acts as a constitutive repressor suppressing biofilm development (10, 11), while SinI antagonizes SinR by binding to it, freeing DNA targets for biofilm matrix gene expression, including exopolysaccharides and extracellular proteins (12). γ-PGA, a biofilm component, is synthesized by a multi-enzyme complex encoded by the *pgs* gene cluster (7, 13).

Plant systemic resistance is categorized into induced systemic resistance (ISR) triggered by beneficial microorganisms and systemic acquired resistance (SAR) induced by pathogenic microorganisms (14). After root colonization, *Bacillus* strains can stimulate plant defense by triggering the induced systemic resistance (ISR) in hosts (15, 16). Microorganisms can produce extracellular polysaccharides, extracellular proteins, and lipopeptides to trigger resistance in plants (16–19). For example, the *B. subtilis* family including *B. velezensis* can activate resistance both on leaves and roots by production of some secondary metabolites such as lipopeptide (20, 21), so they can trigger plant immunity to afford the plant protection against a broad range of pathogen infection (5, 22).

*Ralstonia solanacearum*, a soil-dwelling pathogen, causes the pervasive bacterial wilt disease around the world (23). Our prior work isolated *B. velezensis* R9 from tobacco's rhizosphere, effective in field control of bacterial wilt. *B. velezensis*, a well-studied PGPR, excels in root colonization and serves as a commercial biocontrol against soil pathogens (12, 21). Herein, we aimed to genetically enhance R9's biofilm formation, root colonization, and plant resistance induction against bacterial wilt. Simultaneously, we also contrasted *sinI* and *sinR* functions in biofilm formation between *B. velezensis* and *B. subtilis*.

## MATERIALS AND METHODS

### Bacterial strains, primers, and reagents

The bacterial strains and primers are listed in Table 1; Table S1, respectively. Nucleotide sequences were determined by Sangon Biotech (Shanghai, China). The enzymes were purchased from Takara Bio (China). Chemicals of analytical grade were supplied by Sinopharm Chemical Reagent (China). *B. velezensis* R9 was from the previous studies (24). *B. subtilis* M6 was isolated from the rhizosphere soil of tobacco plants grown in Enshi City, Hubei Province, China.

**TABLE 1** Bacterial strains used in this study

| Strain | Characteristic | Source |
|---|---|---|
| *Bacillus subtilis* M6 | Wild type | Stored in this lab |
| *Bacillus velezensis* R9 | Wild type | Stored in this lab |
| R9Δ*sinI* | *sinI* knockout strain of R9 | Stored in this lab |
| R9Δ*sinR* | *sinR* knockout strain of R9 | Stored in this lab |
| R9Δ*sinI*/T2-*BVsinI* | R9Δ*sinI* compensated with *sinI* from R9 | This study |
| R9Δ*sinI*/T2-*BSsinI* | R9Δ*sinI* compensated with *sinI* from M6 | This study |
| R9Δ*sinI*/T2-*BSsinR* | R9Δ*sinI* compensated with *sinR* from M6 | This study |
| R9Δ*sinR*/T2-*BVsinR* | R9Δ*sinR* compensated with *sinR* from R9 | This study |
| R9Δ*sinR*/T2-*BSsinR* | R9Δ*sinR* compensated with *sinR* from M6 | This study |
| R9Δ*sinR*/T2-*BSsinI* | R9Δ*sinR* compensated with *sinI* from M6 | This study |
| M6Δ*sinI* | *sinI* knockout strain of M6 | This study |
| M6Δ*sinR* | *sinR* knockout strain of M6 | This study |
| M6Δ*sinI*/T2-*BSsinI* | M6Δ*sinI* compensated with *sinI* from M6 | This study |
| M6Δ*sinI*/T2-*BVsinI* | M6Δ*sinI* compensated with *sinI* from R9 | This study |
| M6Δ*sinI*/T2-*BVsinR* | M6Δ*sinI* compensated with *sinR* from R9 | This study |
| M6Δ*sinR*/T2-*BSsinR* | M6Δ*sinR* compensated with *sinR* from M6 | This study |
| M6Δ*sinR*/T2-*BVsinR* | M6Δ*sinR* compensated with *sinR* from R9 | This study |
| M6Δ*sinR*/T2-*BVsinI* | M6Δ*sinR* compensated with *sinI* from R9 | This study |

## Construction of knockout, complementary, and heterologous expression strains

*B. velezensis* R9 knockout strains including R9Δ*sinI* and R9Δ*sinR* were constructed previously (24). *B. subtilis* M6 knockout strains, such as M6Δ*sinI* and M6Δ*sinR*, were constructed according to a previous study (25). Two homologous arms (~500 bp) to the 5′ and 3′ coding regions of the targeted genes (*sinR* or *sinI*) were amplified from *B. subtilis* M6 by PCR with primers listed in Table S1, ligated by splicing with Overlapping Extension PCR (SOE-PCR), and then subcloned into the vector T2(2)-ori with a temperature-sensitive replicon to promote single crossover. The resulting plasmids were used to transform M6 by electroporation, then the transformants were selected by kanamycin resistance and verified by PCR. The PCR-selected transformants were cultured in LB medium with kanamycin (20 µg/mL) at 45°C for 8 h to promote the first crossover, then the mutants with a single crossover were selected by PCR. The strain that had been verified with a successful single crossover was selected and inoculated into an LB liquid medium without kanamycin. The culture was incubated at 37°C with shaking at 200 rpm for 8 h. Following incubation, the culture was diluted appropriately and plated onto an LB solid medium. After 24–48 h of incubation at 37°C, when bacterial colonies had formed, individual colonies were picked and streaked onto both LB agar plates and LB agar plates containing kanamycin. After that, kanamycin-sensitive colonies were screened and verified by colony PCR using the related LF and RR primers listed in Table S1 to confirm the double crossover in cells. Positive colonies were selected for further sequencing analysis to confirm the genetic modification. Complementary and heterologous expression strains were constructed as follows. The plasmids were constructed for expression of *sinI* and *sinR*, respectively. Briefly, the genes including BS*sinI* (*sinI* from *B. subtilis* M6), BV*sinI* (*sinI* from *B. velezensis* R9), BS*sinR* (*sinR* from *B. subtilis* M6), and BV*sinR* (*sinR* from *B. velezensis* R9) with their promoters and terminators were, respectively, amplified from the genomic DNA of M6 or R9 by PCR with the primers listed in Table S1, cloned into the T2(2)-ori plasmid joined by *BamH* I and *Xba* I restriction sites, then verified by PCR. The PCR products were sequenced to confirm the correct insertion of *sinI* or *sinR* in the plasmids, then used for transformation of the related hosts including M6, R9, M6Δ*sinI*, R9Δ*sinI*, M6Δ*sinR*, and R9Δ*sinR* by electroporation, respectively (Table 1). The positive transformants were selected by kanamycin resistance, verified by PCR, and sequenced as described above (26).

## Analysis of biofilm formation and quantifying EPS, extracellular proteins, and γ-PGA in biofilm

Strains were cultured on LB agar plates. The colony morphology was observed under a microscope. Robust pellicle (floating biofilm) was determined in 24-well cell plates. Briefly, the single colony was grown in LB medium at 37℃ overnight, and then 20 μL of broth was inoculated in 2 mL MSgg medium in each well. The plates were incubated at 28℃ for 48 h until floating biofilm formed (24). Biofilm was quantitatively analyzed by crystal violet staining (27). EPS within the biofilm were quantified using the phenol-sulfate colorimetric method. Extracellular proteins in the biofilm were detected by Protein Quantification Kit (Bradford Assay). γ-PGA in biofilm was detected by the cetyltrimethy-lammonium bromide (CTAB) method (28).

## Determining cell growth and sporulation

A single colony of the selected strain was inoculated into 5 mL of LB liquid medium and incubated overnight at 37℃ with shaking at 200 rpm. The next day, 2 μL of the overnight culture was transferred to each well of a 96-well plate containing 200 μL of fresh LB liquid medium per well. The plates were then incubated at 37℃ for 48 h in an automatic growth curve analyzer (Bioscreen C MBR; Oy Growth Curves Ab Ltd., Finland) to measure the growth curves. Thereafter, the average $OD_{600}$ of six wells was used for making growth curves of different strains. Sporulation was detected via crystal violet staining at 48 h (29).

## Detecting reactive oxygen species in plants

$H_2O_2$ is a kind of reactive oxygen species (ROS). A pot experiment was performed to determine the effect of knockout strains on triggering $H_2O_2$ accumulation in tobacco leaves. Each pot was transplanted with a single tobacco seedling (*Nicotiana tabacum* cv. Yunyan 87) containing seven leaves. The pots were randomly assigned to four experimental groups, with 10 seedlings per group. Each seedling was irrigated with 5 mL of a cell suspension from different bacterial strains, each at a concentration of $5 \times 10^9$ CFU/mL. Seedlings irrigated with water alone served as the control group. Three days after the initial treatment, each seedling was irrigated again with 5 mL of *R. solanacearum* suspension at a concentration of $2.5 \times 10^6$ CFU/mL. After that, leaves were stained with diaminobenzidine (DAB) and observed by a dissecting microscope to detect the accumulation of $H_2O_2$ (30).

$H_2O_2$ was also quantitatively determined in plants. The leaves were homogenized in 1 mL water and centrifuged at $12,000 \times g$ for 1 min. A 300 μL supernatant was mixed with 2 mL xylenol orange solution and incubated at room temperature for 30 min. The $OD_{560}$ value was detected by spectrophotometer. $H_2O_2$ content is represented by the ratio of the $OD_{560}$ value of the sample to the maximum $OD_{560}$ value in the test (31).

## qRT-PCR

As described above, tobacco plants were treated with the cell suspension ($5 \times 10^9$ CFU/mL) of R9, R9Δ*sinI*, and R9Δ*sinR*, respectively, and the plants only irrigated with water were used as control. After treatment for 12, 24, and 48 h, the leaves (0.1 g per seedling) with the same leaf arrangement were collected for isolating RNA by RNeasy Mini Kit (Qiagen, Germany). cDNA was produced by reverse transcription with 1 μg RNA, iScript Select cDNA Synthesis Kit, and random oligonucleotide primers. qRT-PCR was performed with cDNA, SsoAdvanced Universal SYBR Green Supermix, and target-specific primers for NPR1, PR-1, PR-5, Coi1, ETR1, and PDF1.2 (Table S2) in CF96 Real-Time System (Bio-Rad, USA) as follows: 95℃ for 5 min, 40 cycles of 95℃ for 10 s, 45℃ for 20 s, and 70℃ for 30 s. The tobacco housekeeping gene Beta-TUB 4 was used as a reference in qRT-PCR. All expression data were normalized to the copy number of Beta-TUB 4 rRNA.

## Detecting activity of plant defense-related enzymes

As described above, the tobacco plants were treated with the cell suspension ($5 \times 10^9$ CFU/mL) of R9, R9$\Delta$sinI, and R9$\Delta$sinR, respectively, and the plants only irrigated with water were used as control. At 0, 12, 24, and 48 h, tobacco leaves (0.25 g per seedling) were collected respectively. Also, the tobacco plants were treated with the cell suspension ($5 \times 10^9$ CFU/mL) of R9, R9$\Delta$sinI, and R9$\Delta$sinR, and the plants only irrigated with water were used as control, then inoculated with *R. solanacearum* after 3 days as that for detection of reactive oxygen species. After that, tobacco leaves (0.25 g per seedling) were collected at 0, 12, 24, and 48 h. The collected tobacco leaves were used to detect the activities of peroxidase (POD), polyphenol oxidase (PPO), and catalase (CAT) according to the methods described previously (32, 33).

## Determining bacterial colonization in rhizosphere soil and root

Strains including R9, R9$\Delta$sinI, and R9$\Delta$sinR were transformed with T2(2)-ori plasmid with kanamycin resistance (30). The transformants were grown in LB medium with 20 µg/mL kanamycin at 37℃ at 180 rpm for 24 h. The culture was centrifuged at $8,000 \times g$ at 4℃ for 10 min. Cell pellets were collected and suspended in water. Tobacco seedlings in pots were randomly divided into three groups with eight seedlings (pots) per group for each time point, which were incubated in a growth chamber at 28℃ with a 16/8 h light/dark regime and 60% relative humidity. Five milliliter of cell suspension ($5 \times 10^9$ CFU/mL) was irrigated onto one tobacco seedling. After 7, 14, 21, 28, and 42 days, the rhizospheric soils were collected from the seedlings, respectively (34). The soils collected from eight plants were mixed as a composite soil sample. One gram of soil samples was suspended in 9 mL of sterilized water and serially diluted. The dilutions were spread on LB plates with 20 µg/mL kanamycin and incubated at 37℃ for 24 h. The colony numbers were counted. Roots were sterilized with 70% ethanol for 40 s and 3% sodium hypochlorite for 5 min, washed with sterilized water three times, and then homogenized and spread on LB plates with 20 µg/mL kanamycin. After incubation at 37℃ for 24 h, the colony numbers were counted. The counted colony number (CFU) was used for indicating the ability of R9, R9$\Delta$sinI, and R9$\Delta$sinR to colonize in tobacco rhizosphere and root as CFU/g soil or root.

## Field experiment and statistical analysis

The field experiment was performed in Enshi State, Hubei Province, China. A block experiment was designed. Four groups were set up including R9, R9$\Delta$sinI, and R9$\Delta$sinR treatment, and control (only treatment with water). Each group included three plots. Sixty seedlings were treated in each plot. Each seedling was irrigated with 200 mL of diluted fermentation at the root. Three plots were set up for each treatment. Biocontrol agents including diluted fermentation of R9, R9$\Delta$sinI, and R9$\Delta$sinR were applied twice at 30 and 60 days after tobacco transplantation. The application of an equal volume of tap water was used as a control. At 30 days after the second application of biocontrol agents, disease incidence, disease severity index, and biocontrol efficiency were investigated. The disease incidence was calculated by the percentage of diseased tobacco in each group. Disease severity was evaluated using a disease score method: 0 = no symptom, 1 = below one-half of tobacco leaves wilted, 3 = one-half to two-thirds of tobacco leaves wilted, 5 = above two-thirds of tobacco leaves wilted, 7 = all leaves wilted, and 9 = stems collapsed or tobaccos died. The disease index was calculated using the formula: Disease index = [$\Sigma$(r × N)/(n × R)] × 100. *r* is the disease severity, *N* is the number of infected tobaccos with a rating of *r*, *n* is the total number of tobaccos tested, and *R* is the value of the highest disease severity in each group. Biocontrol efficiency (%) = (Disease index in control − Disease index in the treatment)/Disease index in control × 100% (35).

   All experiments were repeated in triplicates. The significant differences were analyzed by Student *t*-test. The significant level was $P < 0.05$ (*), and the extremely significant level was $P < 0.01$ (**).

## RESULTS

### Phenotype and biofilm component of R9 knockout strains

The R9ΔsinI colony exhibited a flatter and drier morphology, distinct from the wild-type R9's characteristic "crater" shape. By contrast, the ΔsinR colony was smooth and sticky, setting it apart from both R9 and R9ΔsinI (Fig. 1A).

A robust biofilm was formed on the surface of R9 culture. Compared to R9, the filamentous fibers of floating biofilm disappeared in R9ΔsinR while becoming more robust in R9ΔsinI (Fig. 1A). Consistently, quantitative analysis showed that the biofilm content of R9ΔsinI was higher than R9 (Fig. 1B). Compared to R9, the deletion of sinI led to a significant increase in EPS production, whereas the knockout of sinR resulted in a significant decrease in EPS levels. The content of extracellular proteins in R9ΔsinR biofilm was significantly lower than that in R9. Conversely, the extracellular protein content in R9ΔsinI was significantly higher than that in R9. The γ-PGA content was significantly decreased in R9ΔsinR compared with R9. The γ-PGA content in R9ΔsinI was similar to R9. Collectively, knockout of sinI led to an increase in EPS and extracellular proteins, while deletion of sinR resulted in a significant decrease of EPS, extracellular proteins, and γ-PGA. Both R9ΔsinR and R9ΔsinI showed more vigorous growth than R9 (Fig. S1A). Interestingly, R9ΔsinR was able to sporulate normally, similar to R9. By contrast, R9ΔsinI could also form spores, but the cells remained connected to each other (Fig. S1B). All strains were capable of colonizing the tobacco rhizosphere ($>10^6$ CFU/g soil). The R9ΔsinR strain showed a decreased colonization ability compared to R9, while R9ΔsinI exhibited a stronger colonization ability (Fig. 1C).

At 7 days after inoculation, the colonization ability of R9ΔsinI in the root was significantly higher compared with R9. However, the colonization ability of R9ΔsinR was significantly lower than R9. At 14 days after inoculation, the cell number of R9 alive in the root reached its peak, while the cell number of R9ΔsinI and R9ΔsinR decreased. This decline is likely due to competition with other soil microorganisms, with the wild-type strain R9 demonstrating superior competitive ability. By day 21 post-inoculation, the cell numbers of R9ΔsinI and R9ΔsinR in the root had reached their highest levels, indicating that both mutants had successfully adapted to the soil environment. However, by day 28, the cell numbers of all strains in the root were lower than their respective peak values, likely due to nutrient depletion in the soil (Fig. 1C). Overall, the root colonization capacity of R9ΔsinI increased while R9ΔsinR decreased compared with R9.

The disease incidence among plants treated with R9 was 55.8% ± 3.3 %, which was significantly lower than that in the control group. Plants treated with R9ΔsinR showed a similar disease incidence to R9 treatment. However, plants treated with R9ΔsinI exhibited a significantly lower disease incidence than with R9 (Fig. 1D).

After 30 days of application of R9, R9ΔsinI, and R9ΔsinR fermentation broth to tobacco plants, it was observed that the plant disease index of the R9-treated group was significantly lower than that of the control group, whereas there was no significant difference between the plant disease index of the R9ΔsinR-treated group and that of the control group. The wild bacterium R9 had an inhibitory effect on bacterial wilt disease and significantly reduced the plant disease index, while R9ΔsinI showed a stronger inhibitory effect on bacterial wilt disease (Fig. 1D). R9ΔsinI had the highest biocontrol efficiency of 81.0% ± 1.4% compared to R9 and R9ΔsinR treatment groups (Fig. 1D).

### R9 knockout strain induces $H_2O_2$, resistance, and enzyme activity in plants

When tobacco seedling roots were treated individually with R9, R9ΔsinI, or R9ΔsinR, no detectable accumulation of $H_2O_2$ was observed. By contrast, $H_2O_2$ accumulation was detected following infection with R. solanacearum after the initial treatment with R9, R9ΔsinI, or R9ΔsinR. $H_2O_2$ accumulation was detected in leaves at 12 h and became stronger at 24 h post-infection (Fig. 2A). In particular, R9ΔsinI induced stronger $H_2O_2$ accumulation than other strains.

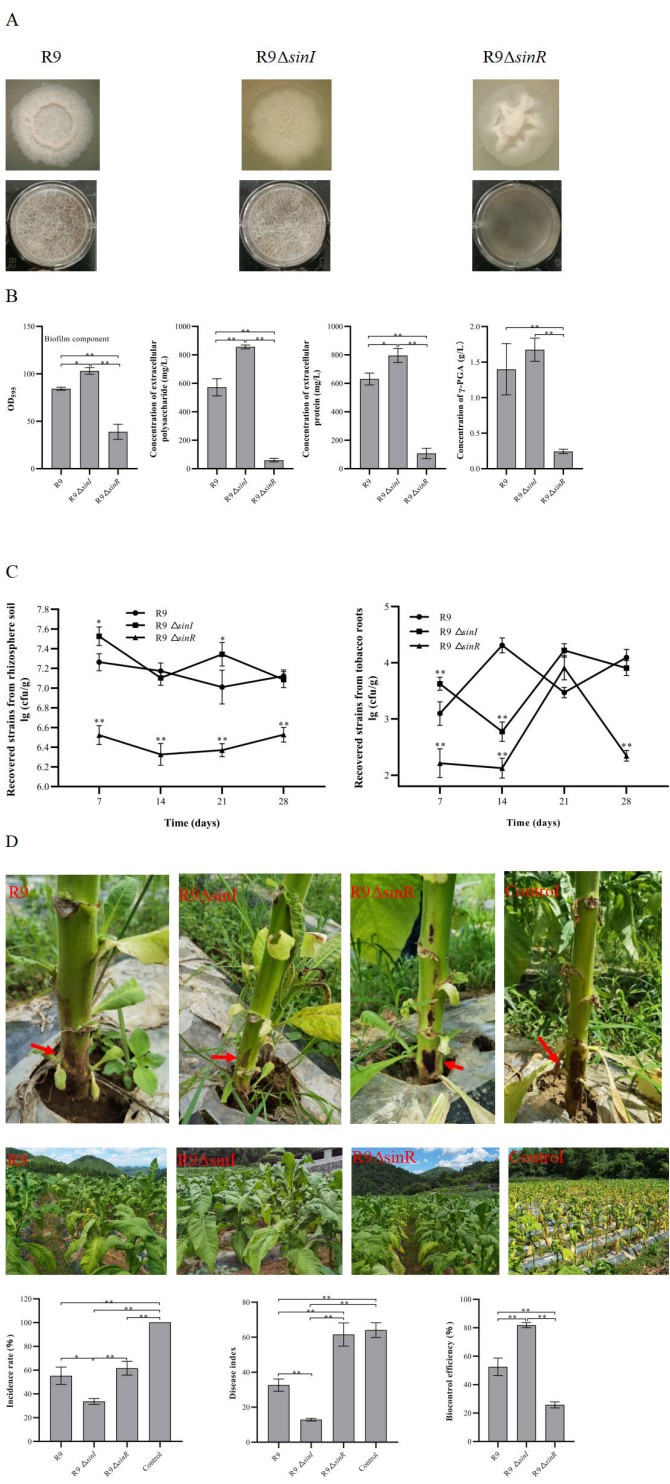

**FIG 1** Phenotypes, biofilm components, colonization, and biocontrol efficiency of R9 mutant strains. (A) Colony morphology and floating pellicle. (B) Quantitative analysis of biofilm and biofilm components, including EPS, extracellular proteins, and γ-PGA. (C) Colonization ability of R9Δ*sinI* and R9Δ*sinR* compared to the wild-type strain R9. (D) Biocontrol effect against tobacco bacterial wilt disease, with the red arrow indicating disease severity. Single asterisk (*P* < 0.05) indicates a significant difference, and double asterisks (*P* < 0.01) indicate a highly significant difference. Error bars represent the standard deviation (SD) of the data.

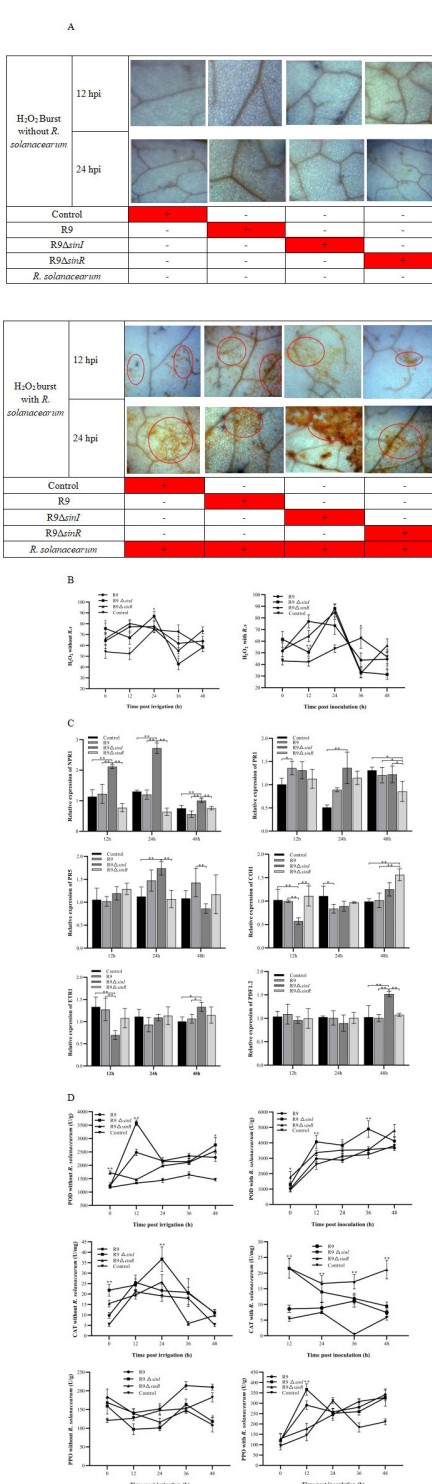

**FIG 2** ROS accumulation, gene transcription, and activity of resistance-related enzymes in leaves post-induction by R9 knockout strains. (A) DAB staining visualizes ROS in leaves treated with R9 knockout strains, subsequently infected with *R. solanacearum*, or uninfected. A plus sign (+) denotes inoculation with the respective strain, while a minus sign (-) signifies no inoculation. (B) Quantitative analysis of $H_2O_2$ in leaves from plants treated with R9 knockout strains, followed by *R. solanacearum* infection or without pathogen infection, compared to R9-treated plants. (C) Transcription levels of plant resistance-related genes in plants treated with R9 knockout strains, compared to the untreated control group. (D) Activities of resistance-related enzymes in plants treated with R9 knockout strains, followed by *R. solanacearum* (Continued on next page)

Fig 2 (Continued)

infection or without pathogen infection, compared to R9-treated plants. Significant differences are indicated by an asterisk ($P < 0.05$), and highly significant differences by double asterisks ($P < 0.01$). Error bars represent the standard deviation (SD) of the measurements.

Quantitative analysis of $H_2O_2$ content revealed that R9Δ*sinI* induced a significantly higher level of $H_2O_2$ at 24 h compared to R9 (Fig. 2B). The $H_2O_2$ level in R9 treatment was maximum at 12 h post-infection, while it reached a maximum value at 24 h for both R9Δ*sinI* and R9Δ*sinR*.

R9 and R9Δ*sinI* upregulated the transcription of *NPR1*, *PR1,* and *PR5*. The transcription level of *NPR1* was significantly higher in the R9Δ*sinI* group than in the R9 group both at 12 h and 24 h. R9Δ*sinI* treatment induced significantly higher transcription levels of *PR1* and *PR5* at 24 h compared with R9 (Fig. 2C). Particularly, R9Δ*sinI* induced stronger plant resistance than other strains.

At 48 h, the transcription level of *COI1* in the plants treated with R9Δ*sinR* was significantly higher than in R9, R9Δ*sinI* treatments, and the control group. The transcription level of *PDF1.2* in plants treated with R9Δ*sinI* increased compared with other groups. The transcription level of *ETR1* in plants treated with R9 and knockout strains was similar to the control group (Fig. 2C).

R9Δ*sinI* induced a higher POD activity than R9. POD activity triggered by R9Δ*sinI* was significantly higher at 12 h compared with R9. After irrigating plants with different strains followed by *R. solanacearum* infection, the POD activity triggered by R9Δ*sinI* was significantly higher than that by R9 at several time points, while the POD activity triggered by R9Δ*sinR* was similar to the R9 treatment (Fig. 2D).

The CAT activity in plants treated with R9 and knockout strains, respectively, achieved a maximum value at 12 h and 24 h. The CAT activity in plants treated with R9Δ*sinI* was significantly higher than R9 at several time points. After irrigating plants with different strains followed by *R. solanacearum* infection, the CAT activity in plants treated with R9Δ*sinI* and R9Δ*sinR* was higher than that in R9 after inoculating the pathogen (Fig. 2D).

The PPO activity in plants treated with knockout strains was lower than that in R9. After irrigating plants with different strains followed by pathogen infection, the PPO activity in the R9 group reached a maximum value at 48 h. The PPO activity of plants treated with R9Δ*sinI* was significantly higher than R9 at 12 h. The enzyme activity of plants treated with R9Δ*sinR* achieved a maximum at 48 h, which was later than the R9 and R9Δ*sinI* groups (Fig. 2D).

## Phenotype and biofilm of *Bacillus subtilis* M6 knockout strains

We compared the functions of *sinI* and *sinR* in regulating biofilm formation in *B. velezensis* and *B. subtilis*. The colony of the M6Δ*sinI* mutant was smooth, which was very different from R9Δ*sinI*. The colony of M6Δ*sinR* was moist (Fig. 3A). The surface of the floating biofilm of M6Δ*sinI* had very little filamentous structure. The contents of biofilm, EPS, and extracellular proteins in M6Δ*sinI* were significantly decreased compared with M6 (Fig. 3B). The function of *sinI* of *B. velezensis* on regulating biofilm formation was in contrast to *B. subtilis*. Interestingly, the biofilm of M6Δ*sinR* did not enhance as expected and formed fewer filaments compared with M6. The deletion of *sinR* in both *B. velezensis* and *B. subtilis* resulted in a significant decrease in the floating biofilm. M6Δ*sinI* grew better than M6, while the growth of M6Δ*sinR* was weaker than M6 (Fig. S1C).

## Compensatory expression of *sinI* in R9Δ*sinI* and M6Δ*sinI*

R9Δ*sinI*/T2-*BVsinI* was obtained by compensating the *sinI* gene of R9 (named *BVsinI*) in R9Δ*sinI*. The colony morphology of R9Δ*sinI*/T2-*BVsinI* differed from both R9Δ*sinI* and R9 (Fig. 4A). The floating biofilm of R9Δ*sinI*/T2-*BVsinI* was reduced compared to R9Δ*sinI*. The contents of biofilm, EPS, and extracellular proteins of R9Δ*sinI*/T2-*BVsinI* were all decreased compared to R9Δ*sinI* (Fig. 4B). Compensation with *BVsinI* inhibited the biofilm

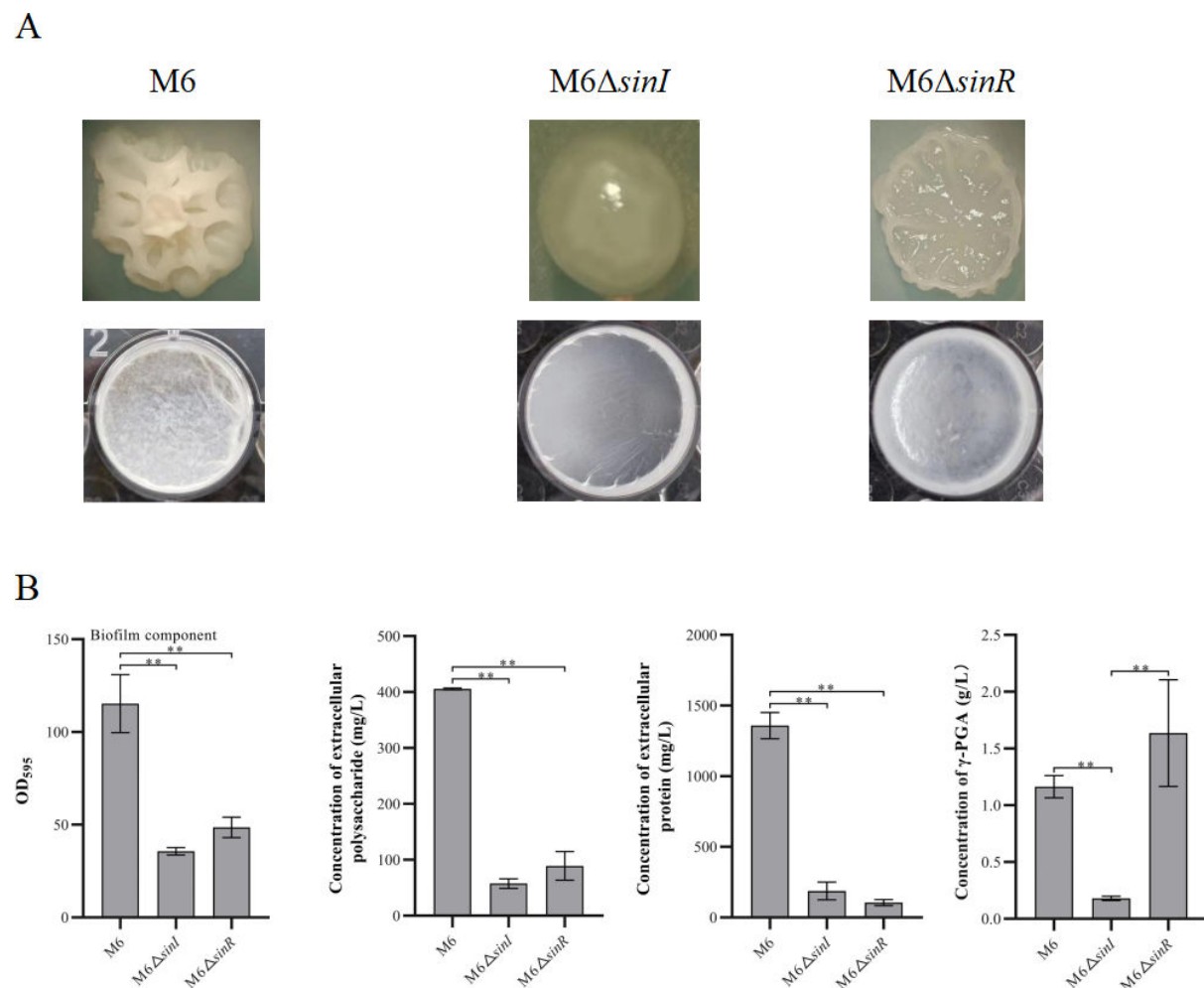

**FIG 3** Phenotype and biofilm components of M6 mutant strains. (A) Colony morphology and floating pellicle. (B) Quantitative analysis of biofilm and biofilm components, including EPS, extracellular proteins, and γ-PGA. Significant differences are indicated by an asterisk ($P < 0.05$), and highly significant differences by double asterisks ($P < 0.01$). The error bars represent the standard deviation (SD) of the measurements.

formation of R9Δ*sinI*, confirming that *sinI* negatively regulated biofilm formation in *B. velezensis*.

R9Δ*sinI*/T2-*BSsinI* was obtained by compensating the *sinI* gene of M6 (named *BSsinI*) in R9Δ*sinI*. The colony morphology of R9Δ*sinI*/T2-*BSsinI* was different from R9Δ*sinI* and R9Δ*sinI*/T2-*BVsinI*. The wrinkle in the colony of R9Δ*sinI*/T2-*BSsinI* was less than R9, R9Δ*sinI,* and R9Δ*sinI*/T2-*BVsinI* (Fig. 4A). The contents of biofilm, EPS, extracellular proteins, and γ-PGA in R9Δ*sinI*/T2-*BSsinI* were significantly lower than in R9, R9Δ*sinI,* and R9Δ*sinI*/T2-*BVsinI* (Fig. 4B). Compensating *BSsinI* suppressed the biofilm formation of R9Δ*sinI*, indicating that *sinI* negatively regulates the biofilm formation of R9 whether *sinI* came from *B. velezensis* or *B. subtilis*.

Moreover, the *sinI* genes from M6 and R9 were used to compensate M6Δ*sinI*, respectively. M6Δ*sinI*/T2-*BSsinI* was obtained by compensating the *BSsinI* gene in M6Δ*sinI*. The colony morphology of M6Δ*sinI*/T2-*BSsinI* was completely different from M6Δ*sinI* but similar to M6 (Fig. 4C). The floating biofilm of M6Δ*sinI*/T2-*BSsinI* was thicker than M6Δ*sinI*. The contents of biofilm, EPS, extracellular proteins, and γ-PGA of M6Δ*sinI*/T2-*BSsinI* were significantly higher than in M6Δ*sinI* (Fig. 4D). M6Δ*sinI*/T2-*BVsinI* was obtained by compensating the *BVsinI* gene in M6Δ*sinI*. The colony morphology of M6Δ*sinI*/T2-*BVsinI* differed from M6Δ*sinI* and M6Δ*sinI*/T2-*BSsinI*, but was similar to R9 with a "crater" shape (Fig. 4C). The contents of biofilm, EPS, extracellular proteins, and γ-PGA of M6Δ*sinI*/T2-*BVsinI* were all higher than M6Δ*sinI* but similar to those in M6Δ*sinI*/T2-*BSsinI* (Fig. 4D). The

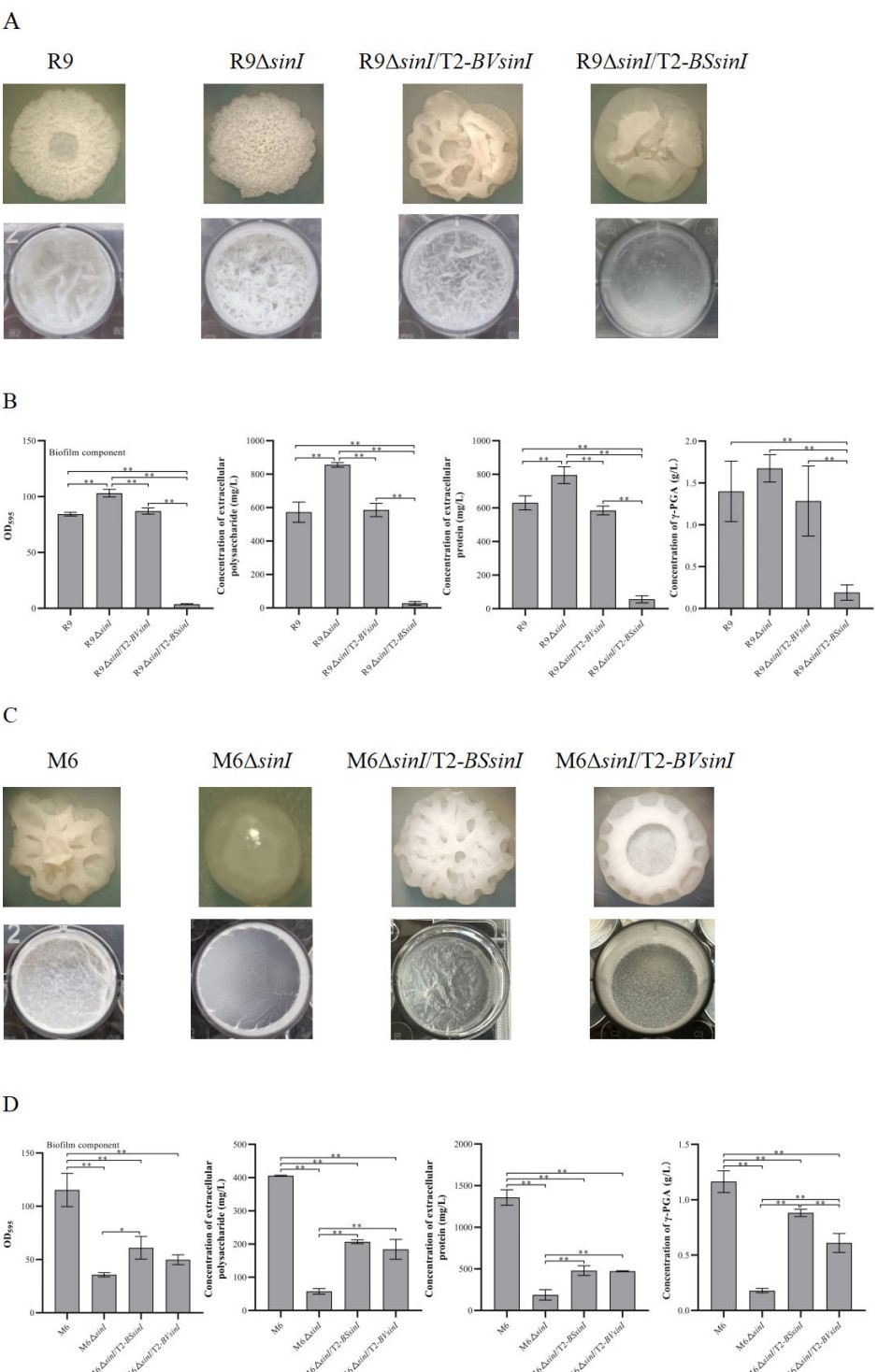

**FIG 4** Biofilm of Δ*sinI* strains complemented with *sinI*. (A) Colony morphology and floating biofilm of R9Δ*sinI* complemented with *sinI* from R9 or M6. (B) Quantitative analysis of biofilm and biofilm components, including EPS, extracellular proteins, and γ-PGA. (C) Colony morphology and floating biofilm of M6Δ*sinI* complemented with *sinI* from R9 or M6. (D) Quantitative analysis of biofilm and biofilm components, including EPS, extracellular proteins, and γ-PGA. Single asterisk (*P* < 0.05) indicates a significant difference, and double asterisks (*P* < 0.01) indicate a highly significant difference. Error bars represent the standard deviation (SD) of the data. M6 and M6Δ*sinI* images are reused from Fig. 3 for ease of comparison.

*sinI* gene positively regulated biofilm formation in M6 whether *sinI* came from *B. subtilis* or *B. velezensis*. The influence of SinI on biofilm formation of *B. velezensis* and *B. subtilis* is different. SinI positively regulates biofilm formation in *B. subtilis* but negatively regulates it in *B. velezensis*.

## Compensatory expression of *sinR* in R9Δ*sinR* and M6Δ*sinR*

R9Δ*sinR*/T2-*BVsinR* was obtained by compensating the *sinR* gene of R9 in R9Δ*sinR*. The colony morphology of R9Δ*sinR*/T2-*BVsinR* was partially restored but still differed from R9Δ*sinR*. The floating biofilm of R9Δ*sinR*/T2-*BVsinR* was restored and was stronger than R9Δ*sinR* (Fig. 5A). The contents of biofilm, EPS, extracellular proteins, and γ-PGA in R9Δ*sinR*/T2-*BVsinR* were significantly higher than in R9Δ*sinR* (Fig. 5B). R9Δ*sinR*/T2-*BSsinR* was obtained by compensating the *sinR* gene of M6 in R9Δ*sinR*. The colony morphology and the floating biofilm were partially restored in R9Δ*sinR*/T2-*BSsinR*. The contents of biofilm, EPS, extracellular proteins, and γ-PGA in R9Δ*sinR*/T2-*BSsinR* were significantly higher than in R9Δ*sinR*. SinR positively regulated the biofilm formation of R9 whether the *sinR* gene came from *B. velezensis* or *B. subtilis*.

M6Δ*sinR*/T2-*BSsinR* was obtained by compensating the *sinR* gene of M6 in M6Δ*sinR*. Interestingly, the inhibitory effect of SinR on biofilm formation became stronger in M6Δ*sinR*/T2-*BSsinR* and was compensated with its *sinR*. The colony of M6Δ*sinR*/T2-*BSsinR* was more viscous and unable to form floating biofilm at 48 h (Fig. 5C). Except for extracellular proteins, the contents of biofilm, EPS, and γ-PGA of M6Δ*sinR*/T2-*BSsinR* were significantly decreased compared to M6Δ*sinR* (Fig. 5D). M6Δ*sinR*/T2-*BVsinR* was obtained by compensating the *sinR* gene of R9 in M6Δ*sinR*. The colony morphology of M6Δ*sinR*/T2-*BVsinR* was similar to M6Δ*sinR*. The floating biofilm of M6Δ*sinR*/T2-*BVsinR* was weaker compared with M6Δ*sinR*. The contents of biofilm and γ-PGA in M6Δ*sinR*/T2-*BVsinR* were significantly decreased compared with M6Δ*sinR*. The contents of EPS and extracellular proteins in the M6Δ*sinR*/T2-*BVsinR* biofilm were similar to M6Δ*sinR*. Compensating with the *sinR* gene inhibited biofilm formation of M6Δ*sinR* whether *sinR* came from *B. velezensis* or *B. subtilis*.

## Heterologous overexpression of *sinI* and *sinR* in mutants Δ*sinI* and Δ*sinR*

The *sinR* gene was heterologously expressed in R9Δ*sinI* and M6Δ*sinI*, respectively. The colony morphology of R9Δ*sinI*/T2-*BSsinR* was flatter than R9Δ*sinI* (Fig. 6A). The floating biofilm of R9Δ*sinI*/T2-*BSsinR* was weakened compared with R9Δ*sinI*. The contents of biofilm, EPS, extracellular proteins, and γ-PGA of R9Δ*sinI*/T2-*BSsinR* were significantly reduced compared with R9Δ*sinI* (Fig. 6B). The heterologous overexpression of the *sinR* gene from *B. subtilis* (named as *BSsinR*) in R9Δ*sinI* might disturb the function of itself SinR (named as *BVsinR*). Differently, the colony morphology of M6Δ*sinI*/T2-*BVsinR* was similar to M6Δ*sinI*, but its colony size was smaller than M6Δ*sinI*. The floating biofilm of M6Δ*sinI*/T2-*BVsinR* was sparser than M6Δ*sinI* (Fig. 6C). The content of biofilm of M6Δ*sinI*/T2-*BVsinR* was significantly lower than in M6Δ*sinI*. However, the contents of EPS, extracellular proteins, and γ-PGA in M6Δ*sinI*/T2-*BVsinR* biofilm were similar to M6Δ*sinI* (Fig. 6D). The heterologous overexpression of the *sinR* gene from *B. velezensis* in M6Δ*sinI* might also have disturbed the function of its own SinR.

The *sinI* gene was heterologously overexpressed in R9Δ*sinR* and M6Δ*sinR*, respectively. The colony of R9Δ*sinR*/T2-*BSsinI* was hollow and kinked and was different from R9Δ*sinR*. The floating biofilm of R9Δ*sinR*/T2-*BSsinI* was weakened compared with R9Δ*sinR* (Fig. 6A). The biofilm content of R9Δ*sinR*/T2-*BSsinI* was significantly lower than R9Δ*sinR*. However, the contents of EPS, extracellular proteins, and γ-PGA in R9Δ*sinR*/T2-*BSsinI* were similar to R9Δ*sinR* (Fig. 6B). The heterologous overexpression of the *sinI* gene from *B. subtilis* in R9Δ*sinR* might have influenced the function of its own SinI.

The colony morphology of M6Δ*sinR*/T2-*BVsinI* was slightly different from M6Δ*sinR*. The floating biofilm of M6Δ*sinR*/T2-*BVsinI* was looser and sparser than M6Δ*sinR* (Fig. 6C). Biofilm content of M6Δ*sinR*/T2-*BVsinI* was significantly decreased compared with

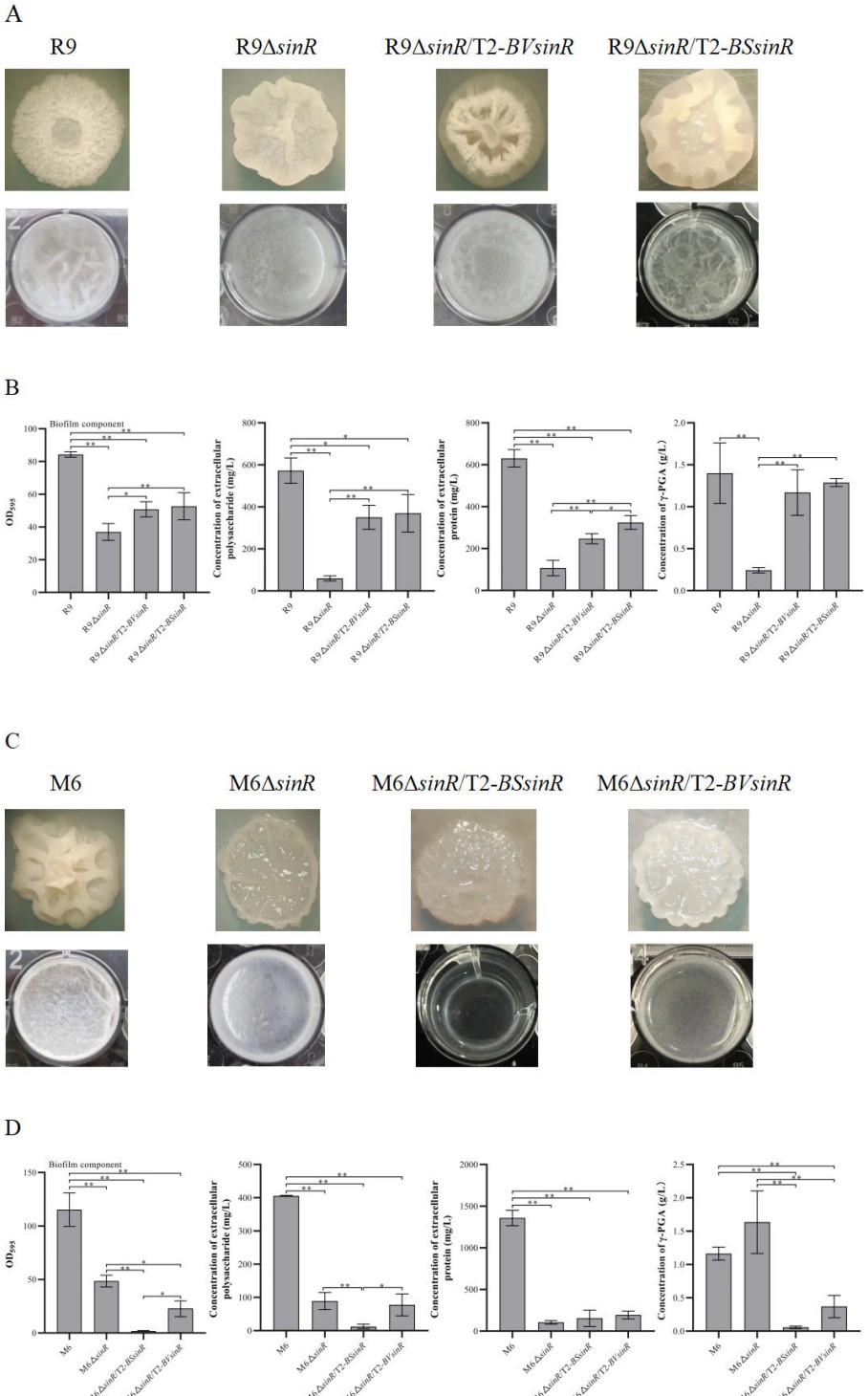

**FIG 5** Biofilm of Δ*sinR* strains complemented with *sinR*. (A) Colony morphology and floating biofilm of R9Δ*sinR* complemented with *sinR* from R9 or M6. (B) Quantitative analysis of biofilm and biofilm components, including EPS, extracellular proteins, and γ-PGA. (C) Colony morphology and floating biofilm of M6Δ*sinR* complemented with *sinR* from R9 or M6. (D) Quantitative analysis of biofilm and biofilm components, including EPS, extracellular proteins, and γ-PGA. Single asterisk (*P* < 0.05) indicates significant difference, and double asterisks (*P* < 0.01) indicate highly significant difference. Error bars represent the standard deviation (SD) of the data. M6, M6Δ*sinR*, and R9 images are reused from Fig. 3 and 4 for ease of comparison.

M6Δ*sinR*. The γ-PGA content of M6Δ*sinR*/T2-*BVsinI* biofilm was significantly lower than M6Δ*sinR*. However, the content of extracellular proteins in M6Δ*sinR*/T2-*BVsinI* biofilm was significantly higher than M6Δ*sinR* (Fig. 6D). The heterologous overexpression of *sinI* from *B. velezensis* in M6Δ*sinR* negatively affected the function of SinI itself.

## DISCUSSION

Biofilm formation is one of the most critical steps during the colonization of biocontrol agents in the plant rhizosphere (36). Here, we constructed *B. velezensis* mutants R9Δ*sinI* and R9Δ*sinR* to boost biofilm and biocontrol efficacy (3, 10, 11). *sinI* knockout elevated EPS and extracellular proteins, whereas *sinR* deletion reduced these components in R9, contrasting *B. subtilis* reports (10). Unlike in *B. subtilis*, in *B. thuringiensis,* SinR does not control EPS manipulators involved in the production of extracellular polysaccharides but is involved in the regulation of lipopeptide biosynthesis, a lipopeptide that is required for biofilm formation (37), which also suggests that there are differences in biofilm regulation in different *Bacillus* species. Despite SinR's antagonistic role in *B. subtilis* (38), *sinR* deletion diminished biofilm in *B. velezensis* R9, weakening its colonization and biocontrol against bacterial wilt. *sinR* compensation from either species could partially restore R9Δ*sinR*'s biofilm and components, confirming SinR's positive regulation in *B. velezensis*. In *Agrobacterium tumefaciens*, SinR, an FNR-type transcriptional regulator, is required for biofilm maturation, and biofilm formation is accelerated upon elevated *sinR* expression (39). SinR deletion in *B. subtilis* M6 similarly decreased biofilm, and compensatory *sinR* expression further reduced it, differing from R9Δ*sinR*. These findings highlight distinct SinR and SinI functions in biofilm regulation between *B. velezensis* and *B. subtilis*.

SinI sequesters SinR, alleviating its biofilm repression, thus positively regulating biofilm formation in *B. subtilis* (5, 6). Here, we found that *sinI* deletion in *B. velezensis* augmented biofilm strength and fiber robustness, diverging from *B. subtilis* findings (10, 38). Biofilm formation is crucial for successful rhizosphere colonization, with stronger biofilms indicating more efficient colonization (40). In this study, the increased levels of extracellular proteins and exopolysaccharides (EPS) in the R9Δ*sinI* biofilm contributed to enhanced fiber formation and attachment (20, 22, 41–43). Xu et al. (4) reported that the Δ*gtaB* mutant of *B. velezensis* SQR9 lost its ability to synthesize EPS, resulting in a deficiency in forming a normal biofilm architecture necessary for efficient rhizosphere colonization. Similarly, Al-Ali et al. (44) found that a mutant of *B. velezensis* FZB42 lacking EPS production (EPS⁻) completely lost its capacity to form biofilms and efficiently colonize tomato rhizospheres. In addition to EPS, extracellular proteins play a significant role in rhizosphere colonization. Nishisaka et al. (45) demonstrated that the *tasA* gene, encoding an important extracellular protein, is essential for stabilizing biofilm membrane dynamics and facilitating cellular adaptation, particularly in plant interactions. γ-PGA has also been identified as a key component of biofilms in Bacillus species. Xue et al. (46) found that secretion of γ-PGA by *B. atrophaeus* NX-12 enhanced root colonization and biocontrol activity against plant pathogens. Thus, the elevated levels of biofilm, extracellular proteins, EPS, and γ-PGA in the R9Δ*sinI* strain collectively enhance its rhizosphere colonization and biocontrol efficacy against tobacco bacterial wilt compared to the wild-type R9 strain. Compensating *sinI* from both species in R9Δ*sinI* markedly reduced biofilm. Despite 78% homology (24), SinI proteins from *B. velezensis* and *B. subtilis* both negatively governed *B. velezensis* biofilm formation.

Deleting *sinI* in *B. subtilis* M6 substantially reduced biofilm, EPS, extracellular proteins, and γ-PGA, underscoring SinI's positive regulation of *B. subtilis* biofilm. *sinI* deletion also altered colony morphology and floating biofilm differently than in *B. velezensis* (6). Complementing M6Δ*sinI* with *sinI* from both species restored biofilm content, confirming SinI's positive role in *B. subtilis* biofilm formation. Thus, SinI positively influences *B. subtilis* biofilm but negatively affects *B. velezensis* biofilm formation, highlighting species-specific regulatory mechanisms.

Heterologous *sinI* overexpression suppressed biofilm formation in R9Δ*sinR* and M6Δ*sinR*. Specifically, overexpressing *sinI* from *B. subtilis* reduced biofilm, EPS,

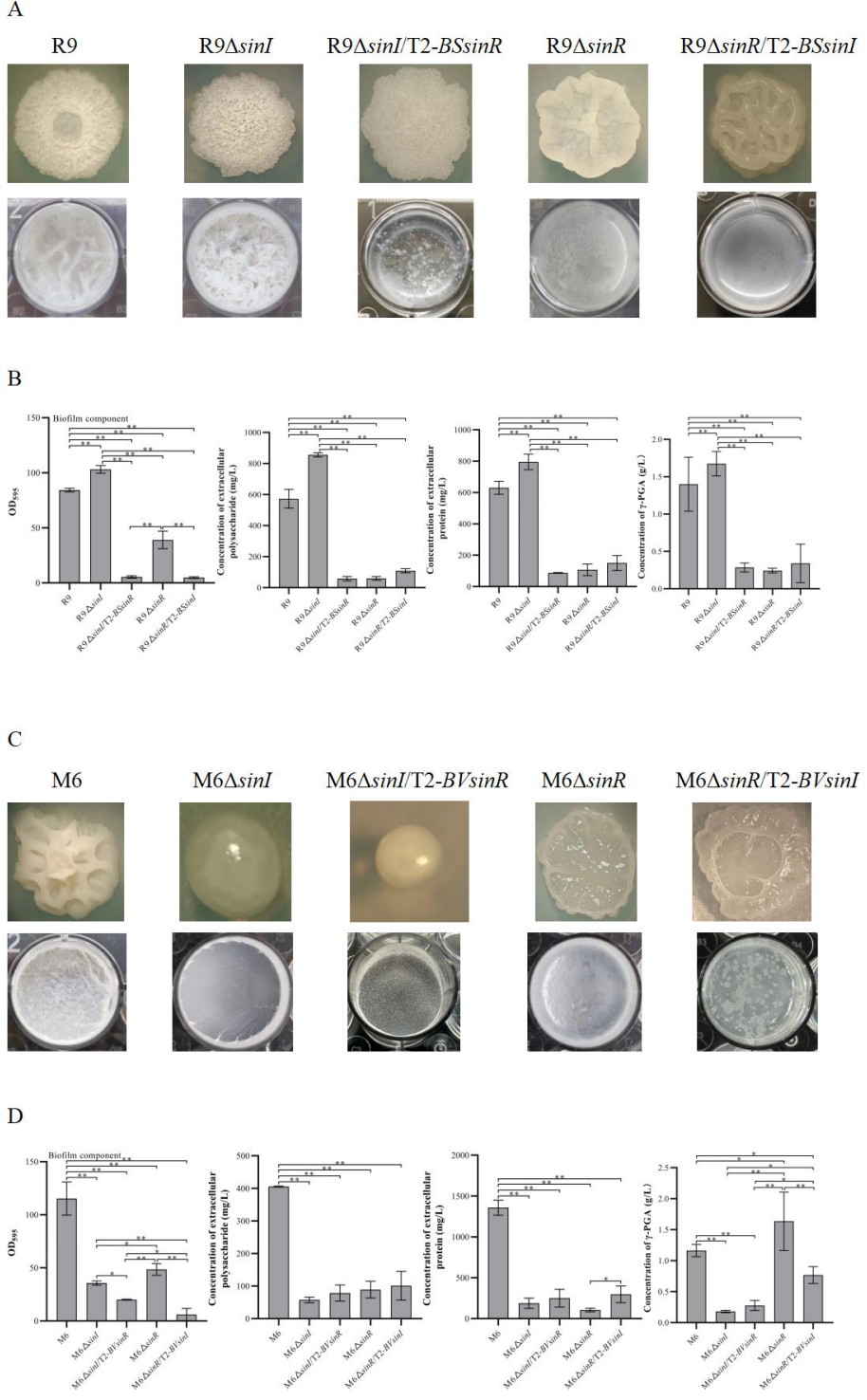

**FIG 6** Biofilm of ΔsinR and ΔsinI strains with heterologous overexpression of sinI and sinR. (A) Colony morphology and floating biofilm of R9ΔsinI and R9ΔsinR with heterologous overexpression of sinR and sinI from M6. (B) Quantitative analysis of biofilm and biofilm components, including EPS, extracellular proteins, and γ-PGA. (C) Colony morphology and floating biofilm of M6ΔsinI and M6ΔsinR with heterologous overexpression of sinR and sinI from R9. (D) Quantitative analysis of biofilm and biofilm components, including EPS, extracellular proteins, and γ-PGA. Single asterisk (P < 0.05) indicates a significant difference, and double asterisks (P < 0.01) indicate a highly significant difference. Error bars represent the standard deviation (SD) of the data. R9, R9ΔsinI, and R9ΔsinR images are reused from Fig. 4 and 5 for ease of comparison. M6, M6ΔsinI, and M6ΔsinR images are reused from Fig. 3 for ease of comparison.

extracellular proteins, and γ-PGA levels in R9ΔsinR. Similarly, overexpression of *sinI* from *B. velezensis* decreased biofilm content in M6ΔsinR. Possibly, heterologous SinI may disrupt native SinI function. Moreover, heterologous *sinR* expression in R9ΔsinI and M6ΔsinI significantly cut biofilm, despite 99% SinR homology between species (24). Overexpressed SinR possibly interferes with endogenous SinR activity, further diminishing biofilm formation in *sinI*-knockout strains.

Induced resistance response is a potent and cost-effective plant defense mechanism against pathogen attacks. Plant growth-promoting rhizobacteria (PGPRs) can trigger induced systemic resistance (ISR) in plants via microbial-associated molecular patterns (MAMPs), such as glycoproteins, peptides, and polysaccharides ( 17–19). These MAMPs are recognized by pattern-recognition receptors (PRRs) located on the plant surface (47). For instance, Nie et al. (48) demonstrated that root-drench application of *B. cereus* significantly reduces disease incidence by activating ISR through MAMP-mediated defense responses. This activation enhances the plant's ability to resist subsequent pathogen attacks.

Reactive oxygen species (ROS), like $H_2O_2$, are signals for inducing resistance in plants (49–51). In an initial experimental phase, tobacco plants were inoculated separately with strains R9, R9ΔsinI, and R9ΔsinR. Observations indicated that none of these strains elicited $H_2O_2$ production during the initial inoculation phase. However, upon subsequent challenge with *R. solanacearum*, the causative agent of bacterial wilt, $H_2O_2$ induction was detected in all cases. Notably, the R9ΔsinI strain demonstrated a significantly heightened induction of $H_2O_2$ compared to the other treatments. Furthermore, treatment with R9ΔsinI treatment was associated with the upregulation of the *NPR1* and *PR1* genes, which are key regulators in the plant's defense signaling pathways, thereby enhancing the plant's immune response (52). Over time, the levels of $H_2O_2$ paralleled the trend observed in the number of viable cells of the mutant strain colonizing the rhizosphere. This correlation suggests that the accumulation of $H_2O_2$ within the plant is linked to bacterial colonization in the rhizosphere, with increased bacterial populations correlating to elevated $H_2O_2$ concentrations. Conversely, R9ΔsinR triggered minimal $H_2O_2$. NPR1, a SA receptor, underpinned R9 and R9ΔsinI-induced resistance via SA signaling (17). The R9ΔsinI strain enhanced the activities of peroxidase (POD), catalase (CAT), and polyphenol oxidase (PPO). This suggests that increased biofilm formation correlates with stronger inter-root colonization ability, enabling plants to produce higher levels of enzymes in response to biotic or abiotic stresses and stimulating the plant's defense response more effectively than treatments with R9 and R9ΔsinR. Field experiment results also demonstrated that the R9ΔsinI treatment group exhibited higher biocontrol efficiency compared to both the R9 and R9ΔsinR treatment groups. In addition, it was observed that the R9ΔsinI treatment group showed significantly greater biocontrol efficiency relative to the R9 and R9ΔsinR groups. Specifically, the R9ΔsinI treatment not only promoted enzyme activity but also provided superior protection against *R. solanacearum*, as evidenced by reduced disease incidence in field trials.

In summary, SinI and SinR function differently in different *Bacillus* species, at least between *B. subtilis* and *B. velezensis*. SinI negatively and SinR positively regulate biofilm formation in *B. velezensis*, contrasting with their roles in *B. subtilis*. Deletion of *sinI* in *B. velezensis* enhances biofilm formation, promotes root colonization, and boosts plant resistance and disease control.

## ACKNOWLEDGMENTS

This work was supported by the Major Tobacco Science and Technology Projects 110202101054 (LS-14) and 027Y2021-017.

## AUTHOR AFFILIATIONS

[1]College of Life Science and Technology, Huazhong Agricultural University, Wuhan, Hubei, China

[2]Enshi Tobacco Technology Center, Enshi, Hubei, China

## AUTHOR ORCIDs

Xiuyun Zhao ⑩ http://orcid.org/0000-0002-6396-2819
Gaofu Qi ⑩ http://orcid.org/0000-0002-7240-339X

## AUTHOR CONTRIBUTIONS

Zhibo Wang, Data curation, Formal analysis, Software | Rui Wang, Methodology, Project administration, Resources | Shilei Jiang, Data curation, Formal analysis | Yuqing Zheng, Data curation, Investigation | Qiankun Jiang, Data curation, Formal analysis | Li Wang, Data curation, Formal analysis | Jun Tan, Methodology | Xiuyun Zhao, Writing – original draft, Writing – review and editing | Gaofu Qi, Resources, Writing – original draft, Writing – review and editing

## ADDITIONAL FILES

The following material is available online.

### Supplemental Material

**Fig. S1 (Spectrum02186-24-s0001.docx).** Growth and sporulation of *sinI* and *sinR* knockout strains.
**Table S1 (Spectrum02186-24-s0002.docx).** Primers used in this study.
**Table S2 (Spectrum02186-24-s0003.docx).** Primers used in qRT-PCR.

### Open Peer Review

**PEER REVIEW HISTORY (review-history.pdf).** An accounting of the reviewer comments and feedback.

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
