## [Reviewer comments · Microbiology Spectrum]

Microbiology Spectrum

SinI and SinR Function differently in Biofilm Formation, Rhizosphere Colonization, and Biocontrol Efficacy between *Bacillus velezensis* and *B. subtilis*

Zhibo Wang, Rui Wang, Shilei Jiang, Yuqing Zheng, Qiankun Jiang, Li Wang, Jun Tan, xiuyun zhao, and Gaofu Qi

Corresponding Author(s): Gaofu Qi, Huazhong Agricultural University

Review Timeline:

Submission Date:	August 31, 2024
Editorial Decision:	October 18, 2024
Revision Received:	January 6, 2025
Accepted:	March 7, 2025

Editor: Yuheng Yang

Reviewer(s): The reviewers have opted to remain anonymous.

Transaction Report:

DOI: <https://doi.org/10.1128/spectrum.02186-24>

Re: Spectrum02186-24 (SinI and SinR Functions differently in Biofilm Formation, Rhizosphere Colonization, and Biocontrol Efficacy between *Bacillus velezensis* and *B. subtilis*)

Dear Prof. Gaofu Qi:

Thank you for the privilege of reviewing your work. Below you will find my comments, instructions from the Spectrum editorial office, and the reviewer comments.

Revision Guidelines

Sincerely,
Yuheng Yang
Editor
Microbiology Spectrum

Reviewer #1 (Comments for the Author):

The manuscript "SinI and SinR functions differently in biofilm formation, rhizosphere colonization, and biocontrol efficacy between *Bacillus velezensis* and *B. subtilis*" by Wang et. al, describes the different effect of SinI and SinR impaired biofilm formation, rhizosphere colonization, induced plant resistance, and bacterial wilt control on *B. velezensis* and *B. subtilis*. The results showed that SinR positively regulate biofilm formation in *B. velezensis* R9, and SinI negative regulate biofilm formation,

root colonization, plant resistance and disease control in *B. velezensis* R9. The study provides novel founding that the research in *B. subtilis* cannot be extrapolated to encompass all *Bacillus* species, at least not *B. velezensis*. The study is interesting, and the results could support conclusions. However, the original ideas and written need to be improved. There were several limitations in the manuscript.

(1) The manuscript used *SinR* and *SinI* to construct the mutants in *B. velezensis* and *B. subtilis*, and evaluated the effect of *SinR* and *SinI* on biofilm formation, rhizosphere colonization, plant resistance inducing, and bacterial wilt control. However, there are other similar function genes that affect biofilm formation of *B. subtilis*? Please explain it.

(2) The results indicated that *SinR* positively regulate biofilm formation, root colonization, plant resistance and disease control in *B. velezensis* R9, which is opposite to the results published in previously study. Please add some similar case in other microbes in discussion to support the founding.

(3) In line 260-271, the results could be supported by Figure 2, such as "The colony of M6 Δ *sinR* was moist and different from R9 Δ *sinR* (Fig. 3A). The surface of the floating 263 biofilm of M6 Δ *sinI* had very little filamentous structure, whereas the biofilm of R9 Δ *sinI* formed many 264 filaments." However, there are no information of R9 in Figure 3. Please check it.

(4) In Figure 1C, the phenotypes of *B. velezensis* and *B. subtilis* on tobacco bacterial wilt should be changed. Single tobacco plant couldn't support the control effect of beneficial bacteria on tobacco bacterial wilt. Please replace photos of occurrence of tobacco bacterial wilt in the whole plot.

(5) In Figure 2C, the results are hard to understand, please change X-axis to inoculated time.

(6) Please revised the introduction and discussion, added more recent research of *Bacillus* on controlling of plant disease, and explain the key founding of why *SinR* act different role in *B. velezensis* and *B. subtilis*.

Reviewer #2 (Comments for the Author):

The authors present the contrasting roles of *SinI* and *SinR* in biofilm formation, rhizosphere colonization, and biocontrol efficacy between *Bacillus velezensis* and *B. subtilis*. The findings highlight species-specific regulatory mechanisms for biofilm formation.

The work is interesting providing some new information relevant for the potential use of biofilm-forming biocontrol agents for in-plant efficacy.

Comments for authors

The introduction is a bit obscured in background information on biofilm development, microbial-associated molecular patterns, and induce systemic resistance (ISR); the authors should provide some context/background on key terms in the study for readers.

From results, the role of the components (Biofilm Content, EPS, γ -PGA, CAT) analyzed were not discussed relative to the in-plant efficacy of any of the test strains.

Other comments are highlighted/provided in the attached manuscript copy.

**SinI and SinR Functions differently in Biofilm Formation, Rhizosphere**

**Colonization, and Biocontrol Efficacy between *Bacillus velezensis* and *B. subtilis***

Zhibo Wang¹, Rui Wang², Shilei Jiang¹, Yuqing Zheng¹, Qiankun Jiang¹, Li Wang¹, Jun Tan², Xiuyun Zhao¹,

Gaofu Qi^{1*}

¹ *College of Life Science and Technology, Huazhong Agricultural University, Wuhan 430070, Hubei Province,*

*China*

² *Enshi Tobacco Technology Center, Enshi City, Hubei Province, China*

* Corresponding author.

Address: Huazhong Agricultural University, Wuhan 430070, China

Tel: +86-15387157411

Email: qigaofu@mail.hzau.edu.cn

**Abstract**

Numerous *Bacillus* species, in particular *B. subtilis* and *B. velezensis*, are usually used as effective
biocontrol agents against plant diseases, leveraging their ability to form biofilms for robust colonization
of the rhizosphere. In *B. subtilis*, SinI positively influences biofilm formation, rhizosphere colonization,
and biocontrol efficacy, whereas SinR has a negative impact. To boost the biocontrol efficacy of *B.*
*velezensis* R9 against tobacco bacterial wilt, we engineered deletion of the *sinI* and *sinR* gene in this
strain, respectively. Contrary to expectations, deleting *sinR* impaired biofilm formation, rhizosphere
colonization, plant resistance induction, and bacterial wilt control. Conversely, the R9 Δ *sinI* strain
showed notably enhanced biofilm, colonization, and biocontrol efficacy relative to both R9 and
R9 Δ *sinR* strains. Complementation of R9 Δ *sinI* with *sinI* and R9 Δ *sinR* with *sinR* confirmed that SinI
negatively and SinR positively regulate biofilm formation in R9, regardless of originating from *B.*
*velezensis* or *B. subtilis*. In contrast, *sinI* knockout in *B. subtilis* M6 caused a marked decline in biofilm
formation, but could be partially reversed by complementary expression of *sinI* whether it was from *B.*
*subtilis* or *B. velezensis*. Conversely, *sinR* knockout in M6 sharply decreased biofilm formation. In
summary, SinI negatively and SinR positively regulates biofilm formation in *B. velezensis*, contrasting
with their roles in *B. subtilis*. Consequently, deleting *sinI*, not *sinR*, in *B. velezensis* enhances biofilm
formation, promoting root colonization, plant resistance, and disease control.

**Keywords:** *Bacillus velezensis*; *sinI*; *sinR*; biofilm; colonization; bacterial wilt disease.

**Importance**

*Bacillus* species, exemplified by *B. subtilis* as a model organism for Gram-positive bacteria, have been
extensively studied, particularly regarding biofilm formation. Biofilms represent a form of quorum sensing in
microbial communities, and the biocontrol efficacy of *Bacillus* species in the rhizosphere, against plant
pathogens, hinges on their biofilm-forming capabilities. In *B. subtilis*, the regulatory proteins SinI and SinR are
known to have opposing functions in biofilm formation, with SinI facilitating and SinR inhibiting biofilm
development. Drawing from this foundational knowledge, we endeavored to knockout the *sinR* gene in *B.*
*velezensis*, a biocontrol bacterium, with the aim of enhancing biofilm formation and, consequently, its
colonization of the rhizosphere and biocontrol efficacy. Unexpectedly, the deletion of *sinR* reduced the
bacterium's proficiency in biofilm formation and its ability to colonize the rhizosphere, resulting in a decrease
in biocontrol effectiveness. On the contrary, the knockout of *sinI* promoted biofilm formation, bolstered the
strain's colonization capacity in the rhizosphere, and thus fortified its biocontrol efficacy. These findings
underscore that SinI and SinR exert divergent, even antithetical effects in *Bacillus* species. Insights gleaned from
*B. subtilis* research cannot be extrapolated to encompass all *Bacillus* species, at least not *B. velezensis*, indicating
the need for species-specific investigations.

**1 Introduction**

*Bacillus* species, such as *B. velezensis* and *B. subtilis*, are Gram-positive, spore-forming bacteria
prevalent in plant rhizospheres. Known as plant growth-promoting rhizobacteria (PGPRs) (Rizzi et al.,
2020), they suppress soil pathogens and foster plant growth by antibiosis, inducing systemic resistance,
and bolstering tolerance to environmental stresses (Rizzi et al., 2019; Xu et al., 2019; She et al., 2020).
Biofilm formation is pivotal for PGPRs' rhizosphere colonization, creating protective multicellular
communities embedded in an extracellular matrix of exopolysaccharides (EPS), proteins, and poly- γ -
glutamic acid (γ -PGA) (Rizzi et al., 2019; Xu et al., 2019; She et al., 2020). This structure fortifies
bacteria against stresses and links robust biofilm capability on roots to enhanced disease control by
PGPRs (Beauregard et al., 2013; Mielich-Süss and Lopez 2015).

In *B. subtilis*, biofilm formation is orchestrated by regulators like SinR and SinI. SinR acts as a
constitutive repressor suppressing biofilm development (Milton et al., 2020; Nishikawa and Kobayashi,
2021), while SinI antagonizes SinR by binding to it, freeing DNA targets for biofilm matrix gene
expression, including EPS and extracellular proteins (Greenwich et al., 2019). γ -PGA, a biofilm
component, is synthesized by a multi-enzyme complex encoded by the *pgs* gene cluster (Arnaouteli et
al., 2021; Chan et al., 2014). After colonizing the rhizosphere, *Bacillus* can produce microbial-
associated molecular patterns (*e.g.*, exopolysaccharides, extracellular proteins, etc.) to induce systemic
resistance (ISR) in plant hosts (Ahn et al., 2007; Pieterse et al., 2014; Jiang et al., 2016; Zeriouh et al.,
2014).

*Ralstonia solanacearum*, a soil-dwelling pathogen, causes the pervasive bacterial wilt disease
around the world (Qi et al., 2019). Our prior work isolated *B. velezensis* R9 from tobacco's rhizosphere,
effective in field control of bacterial wilt. *B. velezensis*, a well-studied PGPR, excels in root
colonization and serves as a commercial biocontrol against soil pathogens (Greenwich et al., 2019;
Stoll et al., 2021). Herein, we aimed to genetically enhance R9's biofilm formation, root colonization,
and plant resistance induction against bacterial wilt. Simultaneously, we also contrasted *sinI* and *sinR*
functions in biofilm formation between *B. velezensis* and *B. subtilis*.

**2 Materials and methods**

**2.1 Bacterial strains, primers and reagents**

The bacterial strains and primers are listed in Table 1 and Table S1, respectively. Nucleotide sequences
were determined by Sangon Biotech (Shanghai, China). The enzymes were purchased from Takara Bio
(China). Chemicals of analytical grade were supplied by Sinopharm Chemical Reagent (China). *B.*
*velezensis* R9 was from the previous studies. M6 was isolated from the rhizosphere soils with planting
tobacco plants located in Enshi City, Hubei Province, China.

**2.2 Construction of knockout, complementary and heterologous-expression strains**

*B. velezensis* R9 knockout strains including R9 Δ *sinI* and R9 Δ *sinR* were constructed previously. *B.*
*subtilis* M6 knockout strains, such as M6 Δ *sinI* and M6 Δ *sinR*, were constructed according to a previous
study. Two homologous arms (~ 500 bp) to the 5' and 3' coding regions of the targeted genes (*sinR* or
*sinI*) were amplified from *B. subtilis* M6 by PCR with primers listed in Table S1, ligated by splicing

with Overlapping Extension PCR (SOE-PCR), and then subcloned into the vector T2(2)-ori with a
temperature-sensitive replicon to promote single crossover. The resulting plasmids were used to
transform M6 by electroporation, then the transformants were selected by Kanamycin resistance, and
verified by PCR. The PCR-selected transformants were cultured in LB medium with Kanamycin (20
96 µg/ml) at 45°C for 8 hours to promote the first crossover, then the mutants with a single crossover were
97 selected by PCR. The selected colonies were picked up and cultured in LB medium at 28°C for 8 h,
then the cells were spread on LB agar plates and replicated on Kanamycin plates for selecting sensitive
colonies. Finally, knockout strains were screened out that had looped out the Kanamycin-resistant gene
by the second crossover, and confirmed by PCR following with nucleotide sequencing of the PCR
products.

Complementary and heterologous-expression strains were constructed as follows. The plasmids
were constructed for expression of *sinI* and *sinR*, respectively. Briefly, the genes including BS*sinI* (*sinI*
from *B. subtilis* M6), BV*sinI* (*sinI* from *B. velezensis* R9), BS*sinR* (*sinR* from *B. subtilis* M6) and
BV*sinR* (*sinR* from *B. velezensis* R9) with their own promoters and terminators were respectively
amplified from the genomic DNA of M6 or R9 by PCR with the primers listed in Table S1, cloned into
the T2(2)-ori plasmid joined by *BamH* I and *Xba* I restriction sites, then verified by PCR. The PCR
products were sequenced to confirm the correct insertion of *sinI* or *sinR* in the plasmids, then used for
transformation of the related hosts including M6, R9, M6Δ*sinI*, R9Δ*sinI*, M6Δ*sinR* and R9Δ*sinR* by

electroporation, respectively (Table 1). The positive transformants were selected by Kanamycin
resistance, verified by PCR and sequenced as described above (Liu et al., 2023).

**2.3 Analysis of biofilm formation and Quantifying EPS, extracellular proteins and γ -PGA in** 113 **biofilm**

Strains were cultured on LB agar plates. The colony morphology was observed under a microscope.
Robust pellicle (floating biofilm) was determined in 24-well cell plates. Briefly, the single colony was
grown in LB medium at 37°C overnight, and then 20 μ L of broth was inoculated in 2 mL MSgg medium
in each well. The plates were incubated at 28°C for 48 h until floating biofilm formed (Zhang et al.,
2022). Biofilm was quantitatively analyzed by crystal violet staining (Branda et al., 2006). EPS within
the biofilm were quantified using the phenol-sulfate colorimetric method. Extracellular proteins in the
biofilm were detected by Protein Quantification Kit (Bradford Assay). γ -PGA in biofilm was detected
by the cetyltrimethylammonium bromide (CTAB) method (Halmschlag et al., 2019).

**2.4 Determining cell growth and sporulation**

The single bacterial colony was grown in LB medium at 37°C overnight, and then 2 μ L broth was
transferred into 200 μ L LB medium in 96-well microplates and incubated at 37°C for 48 h. The growth
curves were determined with an automatic growth curve analyzer (Bioscreen Cpro, OY Growth
Curves, Finland) under the directions of manual. Thereafter, the average of OD₆₀₀ of six wells were
used for making growth curves of different strains. Sporulation was detected via crystal violet staining
at 48 h (Chen et al., 2020a).

**2.5 Detecting reactive oxygen species in plants**

H₂O₂ is a kind of reactive oxygen species (ROS). Pot experiment was performed to determine the effect
of knockout strains on triggering H₂O₂ accumulation in tobacco leaves. One tobacco seedling
(*Nicotiana tabacum* cv. Yunyan 87) with 7 leaves was transplanted in each pot. The pots were
randomly divided into 4 groups. Each group included 10 seedlings. A 5 mL of cell suspension of
different strains (5×10^9 CFU/mL) was irrigated to each seedling. Seedlings only irrigated with water
were used as controls. Three days later, 5 mL *R. solanacearum* suspension (2.5×10^6 CFU mL⁻¹) was
irrigated to each seedling. After that, leaves were stained with diaminobenzidine (DAB) and observed
by a dissecting microscope to detect the accumulation of H₂O₂ (Peng et al., 2019).

H₂O₂ was also quantitatively determined in plants. The leaves were homogenized in 1 mL water
and centrifuged at 12,000 g for 1 min. A 300 µL supernatant was mixed with 2 mL xylenol orange
solution and incubated at room temperature for 30 min. The OD₅₆₀ value was detected by
spectrophotometer. H₂O₂ content is represented by the ratio of the OD₅₆₀ value of the sample to the
maximum OD₅₆₀ value in the test (Choi and Hwang, 2007).

**2.6 qRT-PCR**

As described above, tobacco plants were treated with the cell suspension (5×10^9 CFU/mL) of R9,
R9Δ*sinI* and R9Δ*sinR*, respectively, and the plants only irrigated with water were used as control. After
treatment for 12, 24 and 48 h, the leaves (0.1 g per seedling) with same leaf arrangement were collected
for isolating RNA by RNeasy Mini Kit (Qiagen, Germany). cDNA was produced by reverse

transcription with 1 µg RNA, iScript Select cDNA Synthesis Kit and random oligonucleotide primers.
qRT-PCR was performed with cDNA, SsoAdvanced Universal SYBR Green Supermix and target-
specific primers for NPR1, PR-1, PR-5, Coi1, ETR1 and PDF1.2 (Table S2) in CF96 Real-Time
System (Bio-Rad, USA) as follows: 95°C for 5 min, 40 cycles of 95°C for 10 s, 45°C for 20 s and 70°C
for 30 s. The tobacco housekeeping gene Beta-TUB 4 was used as a reference in qRT-PCR. All
expression data were normalized to the copy number of Beta-TUB 4 rRNA.

**2.7 Detecting activity of plant defense-related enzymes**

As described as above, the tobacco plants were treated with the cell suspension (5×10^9 CFU/mL) of
R9, R9 Δ *sinI* and R9 Δ *sinR*, respectively, and the plants only irrigated with water were used as control.
At 0, 12, 24 and 48 h, tobacco leaves (0.25 g per seedling) were collected respectively. Also, the
tobacco plants were treated with R9, R9 Δ *sinI* and R9 Δ *sinR* as above, then inoculated with
*R. solanacearum* after 3 days as that for detection of reactive oxygen species. After that, tobacco leaves
(0.25 g per seedling) were collected at 0, 12, 24 and 48 h. The collected tobacco leaves were used for
detecting the activities of peroxidase (POD), polyphenol oxidase (PPO) and catalase (CAT) according
to the methods described previously (Soffan et al., 2014; Song and Wang, 2011).

**2.8 Determining bacterial colonization in rhizosphere soil and root**

Strains including R9, R9 Δ *sinI* and R9 Δ *sinR* were transformed with T2(2)-ori plasmid with kanamycin
resistance (Peng et al., 2019). The transformants were grown in LB medium with 20 µg/mL kanamycin
at 37°C at 180 rpm for 24 h. The culture was centrifuged at 8,000 g at 4°C for 10 min. Cell pellets were

collected and suspended in water. Tobacco seedlings in pots were randomly divided into 3 groups with
8 seedlings (pots) per group for each time point, which were incubated in a growth chamber at 28 °C
with a 16/8 h light/dark regime and 60% relative humidity. Five milliliters of cell suspension (5×10^9
CFU/mL) was irrigated onto one tobacco seedling. After 7, 14, 21, 28, and 42 days, the rhizospheric
soil were collected from the seedlings, respectively (Chen et al., 2020b). The soils collected from 8
plants were mixed as a composite soil sample. One gram of soil samples was suspended in 9 mL of
sterilized water and serially diluted. The dilutions were spread on LB plates with 20 µg/mL kanamycin
and incubated at 37°C for 24 h. The colony numbers were counted. Roots were sterilized with 70%
ethanol for 40 s and 3% sodium hypochlorite for 5 min, washed with sterilized water 3 times, and then
homogenized and spread on LB plates with 20 µg/mL kanamycin. After incubation at 37°C for 24 h,
the colony numbers were counted. The counted colony number (CFU) was used for indicating the
ability of R9, R9 Δ *sinI* and R9 Δ *sinR* to colonize in tobacco rhizosphere and root as CFU/g soil or root.

**2.9 Field experiment and Statistical analysis**

The field experiment was performed in Enshi State, Hubei Province, China. A block experiment was
designed. Four groups were set up including R9, R9 Δ *sinI* and R9 Δ *sinR* treatment, and control (only
treatment with water). Each group included 3 plots. Two hundred milliliters of diluted fermentation (5
183 \times) was irrigated per seedling. Sixty seedlings were treated in each plot. Three plots were set up for each
184 treatment. Biocontrol agents including diluted fermentation of R9, R9 Δ *sinI* and R9 Δ *sinR* were applied
twice at 30 and 60 days after tobacco transplantation. The application of an equal volume of tap water

was used as a control. At 30 days after the second application of biocontrol agents, disease incidence,
disease severity index and biocontrol efficiency were investigated according to the methods reported
previously (Wang et al., 2017).

All experiments were repeated in triplicates. The significant differences were analyzed by Student *t* -
test. The significant level was $p < 0.05$ (*), and the extremely significant level was $p < 0.01$ (**).

**3 Results**

**3.1 Phenotype and biofilm component of R9 knockout strains**

The R9 Δ *sinI* colony exhibited a flatter and drier morphology, distinct from the wild-type R9's
characteristic "crater" shape. In contrast, the Δ *sinR* colony was smooth and sticky, setting it apart from
both R9 and R9 Δ *sinI* (Fig. 1A).

A robust biofilm was formed on the surface of R9 culture. Compared to R9, the filamentous fibers
of floating biofilm disappeared in R9 Δ *sinR* while becoming more robust in R9 Δ *sinI* (Fig. 1A).
Consistently, quantitative analysis showed that the biofilm content of R9 Δ *sinI* was higher than R9 (Fig.
1B). Deletion of *sinI* resulted in a significant increase while knockout of *sinR* led to a significant
decrease of EPS when compared with R9. The content of extracellular proteins in R9 Δ *sinR* biofilm
was significantly lower than that in R9. Conversely, the extracellular protein content in R9 Δ *sinI* was
significantly higher than that in R9. The γ -PGA content was significantly decreased in R9 Δ *sinR*
compared with R9. The γ -PGA content in R9 Δ *sinI* was similar to R9. Collectively, knockout of *sinI*

led to an increase of EPS and extracellular proteins, while deletion of *sinR* resulted in a significant
decrease of EPS, extracellular proteins and γ -PGA. Both R9 Δ *sinR* and R9 Δ *sinI* showed a more
vigorous growth than R9 (Fig. S1A). R9 Δ *sinR* could sporulate like R9. R9 Δ *sinI* could also form spores,
but cells were linked together (Fig. S1B).

All strains were capable of colonizing the tobacco rhizosphere ($>10^6$ CFU/g soil). The R9 Δ *sinR*
strain showed a decreased colonization ability compared to R9, while R9 Δ *sinI* exhibited a stronger
colonization ability (Fig. 1C).

At seven days after inoculation, the colonization ability of R9 Δ *sinI* in the root was significantly
higher compared with R9. However, the colonization ability of R9 Δ *sinR* was significantly lower than
R9. The cell number of R9 alive in the root was maximum at 14 days post-inoculation, while the cell
number of R9 Δ *sinI* and R9 Δ *sinR* in the root was maximum at 21 days post-inoculation. The maximum
cell number of R9 Δ *sinI* in the root was higher than R9. The maximum cell number of R9 Δ *sinR* in the
root was lower than R9 (Fig. 1C). Overall, the root colonization capacity of R9 Δ *sinI* increased while
R9 Δ *sinR* decreased compared with R9.

The disease incidence among plants treated with R9 was 55.8 ± 3.3 %, which was significantly
lower than that in the control group. Plants treated with R9 Δ *sinR* showed a similar disease incidence
to R9 treatment. However, plants treated with R9 Δ *sinI* exhibited a significantly lower disease incidence
than with R9 (Fig. 1D).

The disease index of plants treated with R9 was significantly lower than the control group. The
disease index of plants treated with R9 Δ *sinR* was significantly higher than R9 treatment, while the
disease index of plants treated with R9 Δ *sinI* was significantly lower than R9 treatment (Fig. 1D). R9
showed a significantly higher biocontrol efficiency on bacterial wilt disease than R9 Δ *sinR*. R9 Δ *sinI*
showed a significantly higher biocontrol efficiency, with a rate of 81.0 ± 1.4 %, compared with R9
(Fig. 1D).

3.2 R9 knockout strain induces the H₂O₂, resistance, and enzyme activity in plants

Irrigating of roots with R9, R9 Δ *sinI* or R9 Δ *sinR* alone could not trigger H₂O₂ accumulation, but
*R. solanacearum* infection induced weak H₂O₂ accumulation. H₂O₂ accumulation was observed when
plants were first treated with knockout strains followed by *R. solanacearum* infection. H₂O₂

[revised manuscript text omitted]
 (Arnaouteli et al., 2016; Milton et al., 2020). Increased extracellular proteins and EPS
in R9 Δ *sinI* biofilm, contributing to fiber formation and attachment (Romero et al., 2014; Diehl et al.,
2018; Cámara-Almirón et al., 2020; Earl et al., 2020; Otto et al., 2020), enhanced its rhizosphere
colonization and tobacco bacterial wilt biocontrol over R9. Compensating *sinI* from both species in
R9 Δ *sinI* markedly reduced biofilm. Despite 78% homology (Zhang et al., 2022), SinI proteins from *B.*
*velezensis* and *B. subtilis* both negatively governed *B. velezensis* biofilm formation.

Deleting *sinI* in *B. subtilis* M6 substantially reduced biofilm, EPS, extracellular proteins, and γ -
PGA, underscoring SinI's positive regulation of *B. subtilis* biofilm. *sinI* deletion also altered colony
morphology and floating biofilm differently than in *B. velezensis* (Mielich-Süss and Lopez, 2015).

Complementing M6 Δ *sinI* with *sinI* from both species restored biofilm content, confirming SinI's
positive role in *B. subtilis* biofilm formation. Thus, SinI positively influences *B. subtilis* biofilm but
negatively affects *B. velezensis* biofilm formation, highlighting species-specific regulatory
mechanisms.

Heterologous *sinI* overexpression suppressed biofilm formation in R9 Δ *sinR* and M6 Δ *sinR*.
Specifically, overexpressing *sinI* from *B. subtilis* reduced biofilm, EPS, extracellular proteins, and γ -
PGA levels in R9 Δ *sinR*. Similarly, overexpression of *sinI* from *B. velezensis* decreased biofilm content
in M6 Δ *sinR*. Possibly, heterologous SinI may disrupt native SinI function. Moreover, heterologous
*sinR* expression in R9 Δ *sinI* and M6 Δ *sinI* significantly cut biofilm, despite 99% SinR homology
between species (Zhang et al., 2022). Overexpressed SinR possibly interferes with endogenous SinR
activity, further diminishing biofilm formation in *sinI*-knockout strains.

PGPRs trigger plant resistance via microbial-associated molecular patterns, including EPS,
proteins, etc (Ahn et al., 2007; Pieterse et al., 2014; Jiang et al., 2016). Reactive oxygen species (ROS),
like H₂O₂, are signals for inducing resistance in plants (Chen et al., 2015; Nassimi and Taheri, 2017;
Rashid et al., 2017). R9 and its mutants alone did not induce H₂O₂ but primed plants for enhanced
pathogen response (Kierul et al., 2015). Post-infection, ROS accumulated, with R9 Δ *sinI* eliciting peak
H₂O₂ and upregulating NPR1 and PR1 genes. Conversely, R9 Δ *sinR* triggered minimal H₂O₂. NPR1, a
SA receptor, underpinned R9 and R9 Δ *sinI*-induced resistance via SA signaling (Ahn et al., 2007).
R9 Δ *sinI* boosted POD, CAT, and PPO enzyme activities, more effectively priming plants than R9 and

R9 Δ *sinR*. Resistance was also influenced by JA/ET pathways, with COI1 and PDF1.2 for JA and ETR1
for ET signaling (Ahn et al., 2007). R9 Δ *sinI*-induced resistance partially relied on ET signaling, while
R9 Δ *sinR*-induced resistance was partly JA-dependent. Further investigation will explore the differing
resistance mechanisms among strains.

In summary, SinI and SinR function different in different *Bacillus* species, at least between *B.*
*subtilis* and *B. velezensis*. SinI negatively and SinR positively regulate biofilm formation in *B.*
*velezensis*, contrasting with their roles in *B. subtilis*. Deleting *sinI*, not *sinR*, in *B. velezensis* enhances
biofilm formation, promoting root colonization, plant resistance, and disease control.

Reference

- Ahn, I.P., Lee, S.W., Suh, S.C. 2007. Rhizobacteria-induced priming in *Arabidopsis* is dependent on
ethylene, jasmonic acid, and NPR1. *Mol. Plant Microbe Interact.* 20: 759-768.
- Arnaouteli, S., Bamford, N.C., Stanley-Wall, N.R., Kovács, Á.T. 2021. *Bacillus subtilis* biofilm
formation and social interactions. *Nat. Rev. Microbiol.* 19(9): 600-614.
- Arnaouteli, S., MacPhee, C.E., Stanley-Wall, N.R. 2016. Just in case it rains: building a hydrophobic
biofilm the *Bacillus subtilis* way. *Curr. Opin. Microbiol.* 34: 7-12.
- Beauregard, P.B., Chai, Y., Vlamakis, H., Losick, R., Kolter, R. 2013. *Bacillus subtilis* biofilm
induction by plant polysaccharides. *Proc. Natl. Acad. Sci. USA* 110(17): E1621-30.
- Branda, S.S., Chu, F., Kearns, D.B., Losick, R., Kolter, R. 2006. A major protein component of the
*Bacillus subtilis* biofilm matrix. *Mol. Microbiol.* 59: 1229-38.

Cámara-Almirón, J., Navarro, Y., Díaz-Martínez, L., Magno-Pérez-Bryan, M.C., Molina-Santiago, C.,
Pearson, J.R., de Vicente, A., Pérez-García, A., Romero, D. 2020. Dual functionality of the amyloid
protein TasA in *Bacillus* physiology and fitness on the phylloplane. *Nat. Commun.* 11(1): 1859.

Chan, J.M., Guttenplan, S.B., Kearns, D.B. 2014. Defects in the flagellar motor increase synthesis of
poly- γ -glutamate in *Bacillus subtilis*. *J. Bacteriol.* 196: 740-53.

Chen, B., Wen, J., Zhao, X., Ding, J., Qi, G. 2020a. Surfactin: A Quorum-Sensing Signal Molecule to
Relieve CCR in *Bacillus amyloliquefaciens*. *Front. Microbiol.* 11: 631.

Chen, S., Qi, G., Ma, G., Zhao, X. 2020b. Biochar amendment controlled bacterial wilt through
changing soil chemical properties and microbial community. *Microbiol. Res.* 231: 126373.

Chen, X., Mou, Y., Ling, J., Wang, N., Wang, X., Hu, J.M. 2015. Cyclic dipeptides produced by fungus
*Eupenicillium brefeldianum* HMP-F96 induced extracellular alkalinization and H₂O₂ production in
tobacco cell suspensions. *World J. Microbiol. Biotechnol.* 31(1): 247-53.

Choi, H.W., Hwang, B.K. 2007. Hydrogen peroxide generation by the pepper extracellular peroxidase
CaPO₂ activates local and systemic cell death and defense response to bacterial pathogens. *Plant*
*Physiol.* 145(3): 890-904.

Diehl, A., Roske, Y., Ball, L., Chowdhury, A., Hiller, M., Molière, N., Kramer, R., Stöppler, D., Worth,
C.L., Schlegel, B., Leidert, M., Cremer, N., Erdmann, N., Lopez, D., Stephanowitz, H., Krause, E., van
Rossum, B.J., Schmieder, P., Heinemann, U., Turgay, K., Akbey, Ü., Oschkinat, H. 2018. Structural
changes of TasA in biofilm formation of *Bacillus subtilis*. *Proc. Natl. Acad. Sci. USA* 115: 3237-3242.

Earl, C., Arnaouteli, S., Bamford, N.C., Porter, M., Sukhodub, T., MacPhee, C.E., Stanley-Wall, N.R.
2020. The majority of the matrix protein TapA is dispensable for *Bacillus subtilis* colony biofilm
architecture. *Mol. Microbiol.* 114(6): 920-933.

Greenwich, J., Reverdy, A., Gozzi, K., Di Cecco, G., Tashjian, T., Godoy-Carter, V., Chai, Y. 2019.
A Decrease in Serine Levels during Growth Transition Triggers Biofilm Formation in *Bacillus subtilis*.
*J. Bacteriol.* 201(15): e00155-19.

Halmschlag, B., Steurer, X., Putri, S.P., Fukusaki, E., Blank, L.M. 2019. Tailor-made poly- γ -glutamic
acid production. *Metab. Eng.* 55:239-248.

Jiang, C.H., Fan, Z.H., Xie, P., Guo, J.H. 2016. *Bacillus cereus* AR156 extracellular polysaccharides
served as a novel micro-associated molecular pattern to induced systemic immunity to *Pst* DC3000 in
*Arabidopsis*. *Front. Microbiol.* 7: 664.

Kierul, K., Voigt, B., Albrecht, D., Chen, X.H., Carvalhais, L.C., Borriss, R. 2015. Influence of root
exudates on the extracellular proteome of the plant growth-promoting bacterium *Bacillus*
*amyloliquefaciens* FZB42. *Microbiology* 161: 131-47.

Liu, N., Sun, H., Tang, Z., Zheng, Y., Qi, G., Zhao, X. 2023. Transcription factor Spo0A regulates the
biosynthesis of difficidin in *Bacillus amyloliquefaciens*. *Microbiol. Spectr.* 11: e0104423.

Liu, Y., Feng, H., Chen, L., Zhang, H., Dong, X., Xiong, Q., Zhang, R. 2020. Root-Secreted Spermine
Binds to *Bacillus amyloliquefaciens* SQR9 Histidine Kinase KinD and Modulates Biofilm Formation.
*Mol. Plant Microbe Interact.* 2020 33(3): 423-432.

Mielich-Süss, B., Lopez, D. 2015. Molecular mechanisms involved in *Bacillus subtilis* biofilm
formation. *Environ. Microbiol.* 17(3): 555-65.

Milton, M.E., Draughn, G.L., Bobay, B.G., Stowe, S.D., Olson, A.L., Feldmann, E.A., Thompson, R.J.,
Myers, K.H., Santoro, M.T., Kearns, D.B., Cavanagh, J. 2020. The Solution Structures and Interaction
of SinR and SinI: Elucidating the Mechanism of Action of the Master Regulator Switch for Biofilm
Formation in *Bacillus subtilis*. *J. Mol. Biol.* 432(2): 343-357.

Nassimi, Z., Taheri, P. 2017. Endophytic fungus *Piriformospora indica* induced systemic resistance
against rice sheath blight via affecting hydrogen peroxide and antioxidants. *Biocontrol Sci. Techn.* 27:
252-267.

Nishikawa, M., Kobayashi, K. 2021. Calcium Prevents Biofilm Dispersion in *Bacillus subtilis*. *J.*
*Bacteriol.* 203(14): e0011421.

Otto, S.B., Martin, M., Schäfer, D., Hartmann, R., Drescher, K., Brix, S., Dragoš, A., Kovács, Á.T.
2020. Privatization of biofilm matrix in structurally heterogeneous biofilms. *mSystems* 5(4): e00425-
20.

Peng, G., Zhao, X., Li, Y., Wang, R., Huang, Y., Qi, G. 2019. Engineering *Bacillus velezensis* with
high production of acetoin primes strong induced systemic resistance in *Arabidopsis thaliana*.
*Microbiol. Res.* 227:126297.

Pieterse, C.M.J., Zamioudis, C., Berendsen, R.L., Weller, D.M., Van Wees, S.C., Bakker, P.A. 2014.

[revised manuscript text omitted]
*" by Wang et. al, describes the different effect of SinI and SinR impaired biofilm formation, rhizosphere colonization, induced plant resistance, and bacterial wilt control on *B. velezensis* and *B. subtilis*. The results showed that SinR positively regulate biofilm formation in *B. velezensis* R9, and SinI negative regulate biofilm formation, root colonization, plant resistance and disease control in *B. velezensis* R9. The study provides novel founding that the research in *B. subtilis* cannot be extrapolated to encompass all *Bacillus* species, at least not *B. velezensis*. The study is interesting, and the results could support conclusions. However, the original ideas and written need to be improved. There were several limitations in the manuscript.

(1) The manuscript used SinR and SinI to construct the mutants in *B. velezensis* and *B. subtilis*, and evaluated the effect of SinR and SinI on biofilm formation, rhizosphere colonization, plant resistance inducing, and bacterial wilt control. However, there are other similar function genes that affect biofilm formation of *B. subtilis*? Please explain it.

Response: Yes, in addition to SinI and SinR there are similar functional genes that affect biofilm formation in *B. subtilis*, such as the transcription factor Spo0A, Spo0A~P represses the repressor protein AbrB, which represses the manipulator (tapA) that controls the components of the extracellular polysaccharides (epsA-O) and proteins that make up the biological periplasmic matrix, thus indirectly promoting biofilm formation. The deterrent protein AbrB acts in a form similar to that of *sinR*, but is able to independently regulate the formation of the biofilm. In addition, the DegS-DegU two-component regulatory system is involved in the regulation of the biofilm by regulating the expression of the BslA protein and the γ -PGA synthase gene *pgsB*. In addition, there is a double negative feedback regulation between SlrR and

SinR. When SlrR levels are high, SinR and SlrR form a dimer, which rescinds the inhibitory effect of SinR and deters the expression of hag (related to bacteriophage motility), which reduces cell motility and promotes biofilm formation. When the level of SlrR is low, it is inhibited by SinR and the synthesis of biofilm is blocked.

(2) The results indicated that SinR positively regulate biofilm formation, root colonization, plant resistance and disease control in *B. velezensis* R9, which is opposite to the results published in previously study. Please add some similar case in other microbes in discussion to support the founding.

Response: Thank you for your comments. Currently there are studies on the biofilm of *Bacillus* mainly using *Bacillus subtilis* as a model strain, while there are few studies on the mechanism of biofilm formation in other *Bacillus* species, and we have added some case studies to support the fact that there are significant differences between different *Bacillus spp.* in regulating biofilms. For example, in *Bacillus thuringiensis*, *sinR* regulates the synthesis of lipopeptides and thus regulates the formation of biofilms, which is different from the function of *sinR* in *Bacillus subtilis*. In *Agrobacterium tumefaciens*, *sinR* can promote biofilm formation due to *B. velezensis* R9 and *Agrobacterium tumefaciens* belong to different genera of bacteria, and we compared their *sinR* sequences and found very low homology.

(3) In line 260-271, the results could be supported by Figure 2, such as "The colony of M6 Δ *sinR* was moist and different from R9 Δ *sinR* (Fig. 3A). The surface of the floating 263 biofilm of M6 Δ *sinI* had very little filamentous structure, whereas the biofilm of R9 Δ *sinI* formed many 264 filaments." However, there are no information of R9 in Figure 3. Please check it.

Response: We appreciate the feedback you have provided, and we have accordingly revised the original manuscript.

(4) In Figure 1C, the phenotypes of *B. velezensis* and *B. subtilis* on tobacco bacterial wilt should be changed. Single tobacco plant couldn't support the control effect of

beneficial bacteria on tobacco bacterial wilt. Please replace photos of occurrence of tobacco bacterial wilt in the whole plot.

Response: Thank you very much for your advice! We have added photos of whole tobacco fields from different treatment groups to the revised version.

(5) In Figure 2C, the results are hard to understand, please change X-axis to inoculated time.

Response: Thank you for your comments, we have made the necessary revisions to the figures.

(6) Please revised the introduction and discussion, added more recent research of *Bacillus* on controlling of plant disease, and explain the key founding of why SinR act different role in *B. velezensis* and *B. subtilis*.

Response: We are very grateful for your suggestion! We have revised the introduction and discussion in the original manuscript, added some recent research, and discussed the biofilm regulatory networks of *B. velezensis* and *B. subtilis*.

Reviewer #2 (Comments for the Author):

The authors present the contrasting roles of SinI and SinR in biofilm formation, rhizosphere colonization, and biocontrol efficacy between *Bacillus velezensis* and *B. subtilis*. The findings highlight species-specific regulatory mechanisms for biofilm formation.

The work is interesting providing some new information relevant for the potential use of biofilm-forming biocontrol agents for in-planta efficacy.

Comments for authors

The introduction is a bit obscured in background information on biofilm development, microbial-associated molecular patterns, and induce systemic resistance (ISR); the authors should provide some context/background on key terms in the study for readers.

Response: Thank you for your suggestion. We have made some additions to the manuscript.

“background information on biofilm development” was supplemented in INTRODUCTION (Line 64 - 73, Page 4).

“microbial-associated molecular patterns, and induce systemic resistance (ISR)” was added in INTRODUCTION (Line 80 - 90, Page 5) and DISCUSSION (Line 440 - 448, Page 24 and 25).

From results, the role of the components (Biofilm Content, EPS, γ -PGA, CAT) analyzed were not discussed relative to the in-planta efficacy of any of the test strains. Other comments are highlighted/provided in the attached manuscript copy.

Response: Thank you for your suggestion. We have revised the manuscript one by one based on the feedback.

“Biofilm Content, EPS, and γ -PGA” were added in the DISCUSSION (Line 406 - 423, Page 22 and 23)

“CAT” was added in the DISCUSSION (Line 463 - 467, Page 25 and 26)

Finally, thank you very much for your other suggestions! We have already made changes in accordance with the comments you made.

Re: Spectrum02186-24R1 (SinI and SinR Function differently in Biofilm Formation, Rhizosphere Colonization, and Biocontrol Efficacy between *Bacillus velezensis* and *B. subtilis*)

Dear Prof. Gaofu Qi:

Your manuscript has been accepted, and I am forwarding it to the ASM production staff for publication. Your paper will first be checked to make sure all elements meet the technical requirements. ASM staff will contact you if anything needs to be revised before copyediting and production can begin. Otherwise, you will be notified when your proofs are ready to be viewed.

Sincerely,
Yuheng Yang
Editor
Microbiology Spectrum